



# Sensitivity of surface temperature to radiative forcing by cirrus and contrails in a radiative-convective model

Ulrich Schumann[1], Bernhard Mayer[2]

[1]Deutsches Zentrum für Luft- und Raumfahrt, Institut für Physik der Atmosphäre, Oberpfaffenhofen, Germany
[2]Ludwig-Maximilians-Universität München, Meteorologisches Institut, Munich, Germany

*Correspondence to*: Ulrich.Schumann@dlr.de

**Abstract.** Earth surface temperature changes induced by added thin cirrus or contrails are investigated with a radiative-convective-diffusive model, basically without climate system changes, with relaxation of the temperature profile by radiation and mixing. The conceptual study shows that the surface temperature sensitivity to cirrus depends strongly on the ratio of the time scales of energy transport by mixing and radiation, where mixing may include turbulent diffusion, convection and transports by the large-scale circulation. The time scales are derived for steady layered heating (ghost-forcing) and for a transient cirrus case. The time scales are shortest at the surface and shorter in the troposphere than in the mid-stratosphere. Heat induced by cirrus in the upper troposphere reaches the surface only for strong vertical mixing. The local surface-temperature sensitivity to adjusted radiative forcing (RF) is larger for the shortwave (SW) than the longwave (LW) cirrus forcing. For weak mixing, cirrus may cool the surface even if the cirrus causes a positive instantaneous or stratosphere-adjusted radiative forcing (RF) at the tropopause. The shorter time scales near the surface indicate a potential for dominant SW surface cooling regionally where cirrus or contrails form, while weak LW warming may dominate at larger distances.

Keywords: cirrus, contrail cirrus, radiative forcing, climate change, surface temperature, radiation transfer, mixing, efficacy

## 1 Introduction

Upper tropospheric ice clouds (cirrus) warm the troposphere by reducing outgoing longwave (LW) terrestrial radiation and cool by enhancing shortwave (SW) solar radiation backscattering (Stephens and Webster, 1981; Liou, 1986; Sinha and Shine, 1994; Chen et al., 2000). For low optical thickness, the net radiative forcing (RF) from cirrus is often positive at top of the atmosphere (TOA) but negative at the surface (Ackerman et al., 1988; Stackhouse and Stephens, 1991; Fu and Liou, 1993; Jensen et al., 1994; Rossow and Zhang, 1995; Meerkötter et al., 1999; Kvalevåg and Myhre, 2007; Dietmüller et al., 2008; Lee et al., 2009b; Allan, 2011; Berry and Mace, 2014; Hong et al., 2016). For well mixed greenhouse gases, a positive RF implies a global warming (Shine et al., 1994; Hansen et al., 1997a). However, cirrus induces a radiative heat source profile which tends to warm the upper troposphere but may cool the surface (Liou, 1986). Skin and near-surface air temperature changes depend on the surface heat budget which includes contributions from latent and sensible heat exchange





with the atmosphere and the ground (land or ocean) in addition to the net radiation budget (Sellers et al., 1997; Lian et al., 2017). Heat induced in the upper troposphere must be transported downwards to contribute to surface warming, e.g. by convective mixing (Manabe and Wetherald, 1967). Hence, the question whether cirrus clouds cool or warm the Earth surface cannot be simply answered from studies of radiative flux changes alone.

The sensitivity of surface temperature to cirrus changes is of relevance with respect to aviation climate impact by contrails (Lee et al., 2009a; Boucher et al., 2013; Lund et al., 2017). Contrails are cirrus clouds induced by aircraft (Schumann and Heymsfield, 2017). Contrail cirrus of significant optical thickness (>0.1) covers about 0.2 - 0.5 % of the Earth (Minnis et al., 2013; Schumann et al., 2015; Bock and Burkhardt, 2016). Early studies expected a regional surface cooling from contrails (Reinking, 1968). Later, a hemispheric atmosphere warming by contrails was derived from models

(Liou et al., 1990). A special report on Global Aviation of the Intergovernmental Panel on Climate Change (IPCC) (Penner et al., 1999) concluded in 1999: "Contrails tend to warm the Earth's surface, similar to high clouds". Observational evidence for contrail-warming is missing because the expected changes are small, not well correlated with contrail cover, and observed changes may have many causes (Minnis, 2005). Contrail RF contributions depend on many contrail and Earth-atmosphere system properties (Meerkötter et al., 1999; Minnis et al., 1999; Myhre and Stordal, 2001; Schumann et al.,

2012). Contrails are composed of relatively small and aspherical ice particles (Gayet et al., 2012). Hence, contrails may favor the albedo cooling over the greenhouse warming effect, in particular for thin and high contrails and cirrus (Fu and Liou, 1993; Strauss et al., 1997; Wyser and Ström, 1998; Zhang et al., 1999; Marquart et al., 2003; Wendisch et al., 2005; Markowicz and Witek, 2011; Bi and Yang, 2017). Contrail contributions to RF at TOA have been derived from observations (Schumann and Graf, 2013; Spangenberg et al., 2013; Vázquez-Navarro et al., 2015). Most traffic occurs during daytime

causing contrails with higher SW fraction. The global mean positive LW and negative SW parts are nearly cancelling each other with a small positive net RF at TOA. Local increases in LW fluxes below contrails are hardly measurable because tropospheric water vapor effectively shields the surface from contrail-induced LW flux changes (Kuhn, 1970). Local reductions in SW fluxes are well observable at the surface (Khvorostyanov and Sassen, 1998; Haywood et al., 2009; Weihs et al., 2015). Contrails form mainly outside convective clouds in the stably stratified upper troposphere at mid-latitudes

(Schumann et al., 2017), with less efficient vertical heat exchange than in the tropics (Wetherald and Manabe, 1975). Contrails occur mainly over land. It is not sure that the heat induced by contrails in the troposphere over land reaches the ocean by horizontal advection and downward mixing before getting lost to space by radiation. Contrails tend to dehydrate the upper troposphere and reduce ambient cirrus (Burkhardt and Kärcher, 2011; Schumann et al., 2015). Hence, contrails may have the potential to cool (Sassen, 1997). On the other hand, the contrail SW forcing may be less negative because of

higher effective albedo (tropospheric system reflectance) in the extratropics than in the tropics (Stephens et al., 2015). The climate sensitivity for regional forcing at mid-latitudes may be larger than for tropical or globally uniform disturbances (Joshi et al., 2003; Shindell and Faluvegi, 2009). LW forcing may be enhanced while SW forcing may be reduced by humidity and low-level cloud changes (Kashimura et al., 2017). Hence, the equilibrium surface temperature change by contrails cannot be simply deduced from an analogy to high clouds.


The global mean equilibrium change of near-surface air temperature is often approximated by $\Delta T_s = \lambda$ RF as a function of the net downward flux change RF at the tropopause and a "climate sensitivity parameter" $\lambda$ (Houghton et al., 1990). $\lambda$ is similar to the planetary temperature sensitivity parameter $\lambda_p$ to changes in solar irradiance (Stephens, 2005), $\lambda_p = [1/(4 \sigma T_s^3)] (T_s/T_p)^3 [dT_s/dT_p]$. Here $\sigma$ is the Stefan–Boltzmann constant, $T_s$ is the surface temperature, and $T_p$ is the effective

temperature of planetary infrared emissions, $\sigma T_p^4 \cong S_0 (1-a)/4$, with solar irradiance $S_0 \cong 1360$ W m$^{-2}$ and Earth albedo a $\cong$ 0.3. Hence, $\lambda_p \cong 0.267$ K W$^{-1}$ m$^2$ for $[dT_s/dT_p] = 1$. The feedback factor $[dT_s/dT_p]$ differs from one depending on the various forcing types (Stephens, 2005; Bony et al., 2006; Stevens and Bony, 2013). Therefore, $\lambda$ is not a universal constant (Forster et al., 1997; Joshi et al., 2003; Stuber et al., 2005). The "efficacy" e= $\lambda_c/\lambda_{CO2}$, i.e., the ratio of climate sensitivities $\lambda_c$ for non-$CO_2$ disturbances and $\lambda_{CO2}$ for $CO_2$ changes, generally differs from one (Hansen et al., 2005). Various alternative RF

definitions have been suggested to improve the link to climate sensitivity (Boucher et al., 2013; Myhre et al., 2013). The instantaneous $RF_i$ is the RF for a fixed atmosphere. The adjusted $RF_a$ is the RF after thermal relaxation of the stratosphere to the disturbance (Houghton et al., 1990; Stuber et al., 2001). The effective $RF_s$ is the RF after adjustment of the atmosphere to disturbances for constant (ocean) surface temperature (Rotstayn and Penner, 2001; Hansen et al., 2002; Shine et al., 2003). Temperature profile disturbances within the atmosphere relax by thermal relaxation with time scales $t_R$ which are, as we will

further discuss below, of order hours to months depending, among others, on altitude, vertical disturbance scales, and mixing (Manabe and Strickler, 1964; Zhu, 1993). Because of large ocean heat capacity and efficient heat exchange between ocean and atmosphere, the relaxation times scales are far smaller than the time scales for reaching climate equilibrium (Hansen et al., 1981).

Since air traffic is projected to continue to increase for many decades, it is important to know the climate impact of

contrails accurately (Lee et al., 2009a). One-dimensional (Strauss et al., 1997) and two-dimensional radiative-convective models (Liou et al., 1990) showed that contrails may have significant climate impacts. The hope was that three-dimensional global circulation atmosphere/ocean models with a suitable contrail model provide reliable estimates of the climate impact from contrails (Ponater et al., 1996). Various models to represent contrail cirrus in atmospheric global circulation models have been developed (Rind et al., 2000; Ponater et al., 2002; Marquart et al., 2003; Rap et al., 2010b; Burkhardt and Kärcher,

2011; Jacobson et al., 2011; Olivié et al., 2012; Chen and Gettelman, 2013; Schumann et al., 2015), with different treatment of traffic, subgrid scale contrail formation and optical properties. Some of these models were run with atmosphere-ocean coupling (Rind et al., 2000; Ponater et al., 2005; Rap et al., 2010a; Huszar et al., 2013; Jacobson et al., 2013). All these model studies suggest a mean global warming from contrails. The contrail climate effects are expensive to compute because they are small compared to the interannual variability ("climate noise") in climate models (Ponater et al., 1996; Hansen et

al., 1997b), so most studies used by factor 10 to 100 increased disturbances. The contrail efficacy has been computed in a few studies, with results varying from 0.3 to 1 for not fully explained reasons (Hansen et al., 2005; Ponater et al., 2005; Rap et al., 2010a). Avoiding warming and enhancing cooling contrails is considered as a potential concept to mitigate aviation



climate impact if such rating is possible (Schumann et al., 2011; Grewe et al., 2017). Hence, an improved understanding of climate sensitivity to contrail cirrus is urgently needed.

In this conceptual study, we investigate changes in temperature from additional thin cirrus or contrails at mid-latitudes in a radiative-convective model. For understanding of fast adjustment processes, the model is run without climate system
changes ("feedbacks") except thermal relaxation by radiation and mixing. The model is run with highly idealized surface conditions (to reduce the number of free parameters), including constant temperature and zero net vertical heat flux at the surface ("adiabatic surface") as bounding extremes. Instead of investigating the approach to equilibrium with ocean coupling, we simulate the equilibrium atmosphere without heat exchange to an underlying compartment. The disturbances considered are small and, hence, change the reference atmosphere only slightly. For this reason the model is run with fixed
dynamical heating, simulating the heat sources, e.g., from horizontal heat advection, as required for a steady-state reference atmosphere (Strauss et al., 1997). The optical properties of cirrus are essential for its radiative forcing (Fu, 1996; Myhre et al., 2009; Yang et al., 2015), but for this study, the cirrus is just a source of SW and LW radiation flux-profile changes with cloud-radiation interaction details of secondary relevance. Also, aerosol effects are not included in this study. The method is described in Section 2. Section 3 presents the results. Section 3.1 shows the responses of an idealized atmosphere to
prescribed heating, so-called "ghost forcing". This part will point out the importance of the vertical distribution of the radiative heat sources and vertical mixing. The thermal response to an added thin cirrus layer, typical for contrail cirrus, is studied in Section 3.2. We separate the temperature responses to SW and LW radiative disturbances by cirrus and refer correspondingly to "SW cirrus" (similar to a dust layer) and "LW cirrus" (similar to a strong greenhouse gas layer). For constant atmosphere, the sum of SW and LW RF from these cirrus versions is the same as the net RF from "normal" cirrus.
This part will show different temperature responses to SW and LW radiative forcing. A study of thermal relaxation times for cirrus will show up some consequences of temporally and spatially variable cirrus. For comparison and for computation of efficacies for cirrus relative to $CO_2$, we run the same simple model for changed $CO_2$. Section 4 discusses implications of the height-dependent thermal relaxation time scales for global warming from regional cirrus clouds, with SW and LW effects getting advected over different spatial scales. Section 4 also discusses the temperature response to cirrus with some climate
system changes (feedbacks), taking the model with adjusted humidity (Manabe and Wetherald, 1967) as an example for temperature-mediated system changes. Here we show that SW and LW efficacies differ not only for the stratosphere-adjusted RF but also for the effective RF. Finally, Section 5 summarizes the findings and presents conclusions.

## 2 Radiative-convective-diffusive mixing model

This study uses a one-dimensional radiative-convective-diffusive model of the atmosphere with prescribed composition and
clouds, following traditional approaches (Möller and Manabe, 1961; Manabe and Strickler, 1964) with turbulent fluxes as in Ramanathan and Coakley (1978). The model is integrated step-wise in time until steady state. It computes the temperature profile T(z,t) versus altitude z and time t as induced by radiative and turbulent heat transports, based on the heat budget:





$$\rho c_p \frac{\partial T}{\partial t} = -\frac{\partial F_R}{\partial z} - \frac{\partial F_T}{\partial z} + Q_o, \quad F_R = F_{SW}^{up} - F_{SW}^{dn} + F_{LW}^{up} - F_{LW}^{dn}, F_T = -\rho c_p \kappa \left(\frac{\partial T}{\partial z} + \Gamma\right). \tag{1}$$

Here, $\rho$ and $c_p$ are air density and isobaric specific heat capacity, $\Gamma$ is a prescribed threshold lapse rate, and $\kappa = \kappa(t,z)$ is a turbulent diffusivity selected to approximate diffusive mixing (constant $\kappa$) or convective adjustment (large $\kappa$ in case of unstable stratification), as explained below. For contrails and for other small disturbances we compute the temperature

change profile $\Delta T(t,z) = T(t,z) - T_0(z)$ in a given reference atmosphere with temperature profile $T_0(z)$, i.e., we run the model with "fixed dynamical heating" $Q_0$. Here, $Q_0$ is the divergence of the total fluxes $F_R + F_T$, so that $\partial T/\partial t = 0$, for $T = T_0$. Fixed dynamical heating is commonly used for stratospheric adjustment (Ramanathan and Dickinson, 1979; Forster et al., 1997; Myhre et al., 1998) but used here also for tropospheric adjustments of the given reference atmosphere to small disturbances (Strauss et al., 1997). Cases with pure radiative equilibrium ($Q_0 = 0$) are discussed also.

The radiative flux $F_R$ is computed with an efficient two-stream solver using libRadtran (Mayer and Kylling, 2005; Emde et al., 2016). Tests with the more accurate discrete ordinate solver DISORT (Stamnes et al., 1998) show flux differences relative to the two-stream solver of the order 10 %, but DISORT takes far more computing time. Radiation absorption by gases ($H_2O$, $CO_2$, $O_3$, etc.) is calculated with correlated-k distributions for SW (~0.2 - 4 µm) and LW radiation (4 - 70 µm) from Fu and Liou (1992). An alternative SW absorption model from Kato et al. (1999) induces flux differences

small compared to those between the two solvers. The model includes a cirrus layer of hexagonal ice crystals with optical properties from Fu (1996) and Fu et al. (1998).

The turbulent flux $F_T$ is approximated as a function of the temperature gradient including the prescribed lapse rate $\Gamma$ and diffusivity $\kappa$ (Ramanathan and Coakley, 1978; Liou and Ou, 1983). $\Gamma$ is included to make sure that an atmosphere under threshold conditions with $dT/dz = -\Gamma$ experiences zero turbulent fluxes. The added $\Gamma$ drops out in the equations for $\Delta T$ for

fixed dynamical heating because the contribution from $\Gamma$ affects also $Q_0$. The diffusivity $\kappa$ is set to zero in the stratosphere and to a constant $\kappa = 100$ m$^2$ s$^{-1}$ in the troposphere for simulation of diffusive mixing in this study. This value turns out to cause strong vertical mixing in the troposphere with time scales $h^2/\kappa$ of the order of a few days depending on vertical scales $h$ of temperature changes and surface boundary condition. Various methods have been used in the past for "convective adjustment", i.e., enforcement of the lapse rate below a given threshold of, e.g., $\Gamma = 6.5$ K km$^{-1}$ (Manabe and Strickler, 1964;

Ramanathan and Coakley, 1978). Here, we increase the diffusivities by the factor 100 $(2/\pi)$ atan$(\gamma)$, with $\gamma = $ max$[0,(\Gamma + dT/dz)/\Gamma_t]$, allowing for a small departure of $-dT/dz$ from the threshold lapse rate $\Gamma$ by setting $\Gamma_t$ to 0.1 K km$^{-1}$. This causes rapid convective adjustment at timescales shorter than one time step (6 h) and avoids spurious numerical oscillations from the on/off behavior of convection near threshold conditions. The method provides a well-defined turbulent flux, avoids iterations, is numerically stable, and conserves thermal energy.

The numerical scheme uses a non-uniform grid in z with model TOA at 60 km with 100 grid cells vertically. High vertical resolution is necessary to resolve the local flux changes caused by thin cirrus. The lowest layer is centered at 25 m,


the highest at 57.5 km, about 0.3 hPa; the grid spacing is $\Delta z = 250$ m between 0.25 and 19 km height. The radiative solver gets the air temperature and composition at grid centers together with the skin surface temperature as input and returns the fluxes at the grid cell boundaries as output. This staggering avoids 2-$\Delta z$-wave artefacts. Diffusive fluxes are computed implicitly with a tridiagonal Gaussian solver based on the temperatures at the next time step. Pressure is recomputed after

each change in temperature as a function of altitude for air as ideal gas assuming hydrostatic equilibrium for given gravitational acceleration and surface pressure (1013 hPa). The tropopause is defined, as common in meteorology, by the lowest grid interface with $dT/dz > -2$ K km$^{-1}$.

Initial conditions prescribe temperature and composition profiles for the mid-latitude summer standard atmosphere without aerosols (Anderson et al., 1986), see Figure 1. The humidity profile is kept constant unless noted otherwise. Surface

albedo ($A = 0.3$) is selected to mimic an average low-level cloud cover, and the solar zenith angle ($\cos(SZA) = 0.25$) is set such that the downward solar direct radiation equals 1/4 of the solar irradiance as in the global mean. Boundary conditions prescribe either fixed (skin) surface temperature or an adiabatic boundary. An adiabatic boundary is implemented by setting $F_R+F_T= 0$ at the surface. This flux is used when computing the heating rate in the lowest model layer. An adiabatic surface implies zero surface heat capacity and zero total flux between the atmosphere above and the compartment below the surface.

This condition also simulates an atmosphere in thermal equilibrium with the lower compartment (ocean, ice, etc.). We consider two variants to determine the skin temperature $T_{skin}$ at the adiabatic surface. $T_{skin}$ is either set equal to the air temperature $T_s$ in the lowest model layer, implying rapid mixing between the surface and the lowest air layer, or $T_{skin}$ is determined from the surface energy budget for given surface albedo $A$ and unit surface emissivity, $\sigma T_{skin}^4 = F_{SW}^{dn}(1-A) + F_{LW}^{dn}$, implying zero turbulent fluxes at the surface. The code runs stably with 6-h time steps for all

applications in this paper.

The atmosphere responses to the radiative heating with changes of temperature and of the related fluxes, see Eq. (1), until the sum of the changed radiative and turbulent fluxes approach a vertically constant value. For constant surface temperature the fluxes stay non-zero. The fluxes are assumed to be positive for z vertically upwards. Positive upward fluxes imply a cooling, negative a warming of the surface. Over an adiabatic surface, the fluxes approach zero at all heights. During

integration, we monitor the net vertical flux at all relevant altitudes (during stratospheric adjustment only in the stratosphere). The integration is performed until the maximum deviation of the flux values from the mean at all these altitudes is <0.3 % of the maximum instantaneous flux value. Approach to equilibrium is accelerated, during the first half of time steps, by adding, e.g., 5 times the mean heating rates in the troposphere and stratosphere to the temperature changes in the respective layers. Here, the mean heating rates result from the differences between the fluxes at top and bottom of the

layer divided by the layer heat capacity. With this method, radiative equilibrium is reached within the given deviation with less than 640 time steps (160 d).

RF is computed from the difference between the net total fluxes at the tropopause (TP) in model solutions with and without the disturbance. The sign of RF is defined such that positive values imply a warming of the Earth-troposphere





system. For fixed dynamical heating, the model solution without disturbance is given by the steady-state initial conditions. The instantaneous (i), stratospheric adjusted (a), and the effective (s) forcing is computed from three model runs with different boundary conditions. $RF_i$ is the flux change for fixed atmosphere; it varies with height. $RF_a$ is the flux change at the TP after the stratosphere temperature has adjusted to the disturbance for fixed troposphere; it is constant throughout the

stratosphere. $RF_i$ and $RF_a$ are computed for fixed skin surface temperature. The effective $RF_s$ is the flux change at the TP after reaching equilibrium in the entire atmosphere with fixed $T_s$. Here, the total flux is vertically constant. Finally the equilibrium response is computed for an adiabatic surface for which the total flux change is zero at all levels.

The method has been tested with the mentioned alternative solvers and molecular absorption models by comparison of the daily mean and time dependent instantaneous SW and LW RF values of a cirrus layer with results from earlier studies

(Meerkötter et al., 1999); see Figure 2 and Figure 3. The dynamical heating $Q_0$ required to keep the mid-latitude summer atmosphere at steady state is shown in Figure 1. On average, the heating rate from $Q_0$ is 1.39 K $d^{-1}$ in the troposphere and -0.062 K $d^{-1}$ in the stratosphere. These values are similar to the net heating rates presented in fig. 22 of Manabe and Möller (1961). For zero dynamical heating, the code reproduces the approach to pure radiative equilibrium in the atmosphere (Manabe and Strickler, 1964), see Figure 4. Because of strong variations of the heating rate with altitude, the transient

solution tends to form temperature kinks in the lower stratosphere. These kinks disappear slowly when reaching equilibrium because of low energy exchange by radiation between neighboring layers, mainly in the 15-μm $CO_2$-band in regions with low $H_2O$ and $O_3$ concentrations (Plass, 1956). Figure 4 also shows that the model simulates convective adjustment similar to Manabe and Strickler (1964), which illustrates the known importance of vertical mixing for the temperature profile. For a doubled $CO_2$ mixing ratio (from 300 to 600 μmol $mol^{-1}$), the model computes a temperature change of 1.1 K without

feedbacks, similar to previous results (Hansen et al., 1981). The radiative-convective equilibrium solutions with a cirrus layer for zero forcing $Q_0$ are shown in Figure 5 and Figure 6. These results are qualitatively similar to those presented below for deviations from the mid-latitude summer atmosphere. Of course, the mid-latitude summer atmosphere is far less convective than the free radiative equilibrium atmosphere.

### 3 Results

#### 3.1 Temperature response to prescribed heating at various altitude levels

In order to understand air temperature responses to heating at various altitudes, we follow the "ghost" forcing concept of Hansen et al. (1997a). The ghost forcing is a prescribed additive flux change causing a constant heating rate in an altitude interval. The heating causes temperature changes until reaching equilibrium in which the changed fluxes balance the ghost forcing. The model is run for fixed climate system except changing temperature and mixing. In contrast to a forcing by an

added cloud or by changed air composition, the ghost forcing does not change the radiative properties of the atmosphere except by temperature changes.



Eleven simulations are performed with a prescribed flux change of 1 W m$^{-2}$. One simulation is run for a flux change in the lowest model layer above the surface, and ten for flux changes in subsequent 100-hPa pressure intervals between the surface and TOA. The imposed change in net flux is zero at the surface, without direct impact on surface heating, and decreases linearly to -1 W m$^{-2}$ within the heated atmosphere interval. Above the heated layer, the flux is constant reflecting a change of the heat budget between the surface and TOA, so that $RF_i = 1$ W m$^{-2}$ at TOA. For an atmosphere in hydrostatic equilibrium with dp $=-\rho$ g dz, the ghost forcing causes a heating rate (rate of temperature change) H$= (\partial T/\partial t)_R = $ g $(\partial F_r/\partial p)/c_p$. Here, H$= 0.0833$ K d$^{-1}$ in the respective 100-hPa intervals, and 0.825 K d$^{-1}$ for the surface ghost forcing. Figure 7 shows, for example, the heating profile for forcing between 600 - 700 hPa. Figure 8 shows the initial and final flux profiles for these cases. We find that the flux in equilibrium over a constant surface temperature is in between the initial instantaneous flux values at the TP and at the surface.

Figure 9 shows the steady-state temperature profiles in response to the 11 ghost forcings and for three different versions of vertical mixing. In the radiative case with zero turbulent fluxes, the temperature change profiles are similar to vertically smoothed heating rates. The profiles follow the local heating with vertical scales that are the smaller, the higher the effective optical depth for infrared radiation (Stephens, 1984; Goody and Yung, 1989). Radiation causes energy exchange between neighboring layers and between the air layers and the surface. The atmosphere and the surface also emit energy directly to space. Even for heating at the surface, the lowest air layer gets warmer than the surface because the warm black surface emits radiation to space in the transparent thermal infrared window between 8 and 13 μm wavelengths while the air layer emission is weak in this spectral range. Because of lower emissivity and lower temperature, the temperature increase required to balance the ghost forcing is far higher in the stratosphere than in the troposphere. Turbulent vertical mixing smoothes the profiles further, as expected. Convective mixing is rather weak for this case because the mid-latitude summer atmosphere is rather stable compared to the tropics, so that convection occurs only in the upper troposphere where the ghost heating causes local instability.

Figure 10a shows the surface temperature change $\Delta T_s$ as a function of the height of the heated layer. $\Delta T_s$ is, of course, maximum for ghost forcing directly at the surface. Its value depends on details of the surface boundary condition. Here we show results assuming perfect mixing between the surface and the lowest air layer with equal skin and air surface temperatures, $\Delta T_s = 0.371$ K. In the alternative, for $T_{skin}$ computed from the local radiation budget, $T_{skin}$ is far higher (by about 13 K for the given albedo and SZA, which is a realistic magnitude (Lian et al., 2017)) and emits energy more efficiently, so that the skin temperature change induced by ghost forcing is smaller, $\Delta T_{skin} = 0.300$ K, and the air surface temperature change is larger, $\Delta T_s = 0.491$ K. Without diffusive mixing in the troposphere (black circles), $\Delta T_s$ decreases with the height of the heated layer. For strong tropospheric mixing ($\kappa =100$ m$^2$ s$^{-1}$, red symbols), $\Delta T_s$ is 0.260 K for surface ghost forcing, and this value stays close to constant within the whole troposphere. For comparison, Hansen et al. (1997a) (their Table 4 and Fig. 8 a) report a vertically nearly constant $\Delta T_s$ for fixed clouds, with $\Delta T_s = 0.288$ K when normalized to the





same forcing. Apparently their model simulated strong vertical mixing. Small differences were to be expected because of, e.g., different atmospheres.

Figure 10c shows the thermal relaxation time scale $t_R = \Delta T/H$ (in units of days) computed from the steady-state layer-mean temperature change $\Delta T$ in the heated layers at various levels and the given heating rate H. For radiative equilibrium

with zero turbulent fluxes, $t_R$ is 0.45 d near the surface (and smaller for thinner surface air layers), 6.6 d in the first 100 hPa layer, 11 d in the upper troposphere, 30 d in the TP region between 100 and 200 hPa, and 23.5 d in the top 100-hPa layer. For layers with 200 hPa depth instead of 100 hPa, the heating response is smoother, causing about 50 % larger time scales. Hence, the sensitivity to layer depth is less than linear (Goody and Yung, 1989). Radiation causes nonlocal energy transfer, different from diffusion processes for which the sensitivity to layer depth would be quadratic. The smaller time scales in the

lowest layers are again a consequence of effective radiation emission via the surface. The relaxation times in the highest layer are lower than in the second highest layer, because of stronger heat loss from the middle atmosphere to space (Zhu, 1993). Turbulence causes additional mixing reducing the layer warming and the related time scales. Mixing in the troposphere also reduces stratospheric time scales by enhanced heat exchange between air layers near the tropopause by radiation, heat exchange within the troposphere by mixing, and enhanced heat loss from the surface to space. With strong

tropospheric vertical mixing, the thermal relaxation times for heating in the troposphere approach a low and vertically constant value of about 3.2 d. For an atmosphere in which the adiabatic surface is replaced by a constant temperature surface, the time scale $t_R$ is zero at the surface; $t_R$ reduces by 34 % in the first 100-hPa layer, and by 12 % in the second layer, with smaller changes at higher levels. In this case, because of combined transport by radiation and mixing, heat has a lower residence time than a passive tracer with similar source location and constant concentration at the Earth surface.

Passive aircraft emissions may well exceed one month atmospheric residence time when emitted into the lower stratosphere (Forster et al., 2003) but reach ground within less than about a week when emitted in the mid troposphere (Danilin et al., 1998).

Figure 10b and e show the adjusted and effective $RF_a$ and $RF_s$ versus the height of the heated layer. $RF_a$ equals $RF_i = 1$ W m$^{-2}$, regardless of the layer height as long as the heated layer is fully below the TP (Hansen et al., 1997a). The ratio

$RF_s/RF_i$ measures the fraction of heat that continues to warm the compartment below the surface after the air temperature has adjusted to the induced heat disturbance. $RF_s/RF_i$ is largest for heating near the surface: 0.804 in the case without diffusive mixing. Hence, about 80 % of the input heat heats the compartment below the Earth surface (e.g., ocean) and 20 % of the heat radiates out to space when the troposphere has reached its higher steady-state temperature. For heating near the TP, about 95 % of the heat leaves to space. For strong vertical mixing, $RF_s/RF_i$ is about 60 % and vertically nearly uniform.

Hence, even with strong mixing, ~40 % of the ghost heating radiates directly to space. Finally, Figure 10d and f show $\lambda_a$ and $\lambda_s$, the sensitivity parameters of $\Delta T_s$ to $RF_a$ and $RF_s$. For heating at the surface, $\lambda_a = 0.371$ K W$^{-1}$ m$^2$ based on equal skin and air surface temperature. It would be 0.291 K W$^{-1}$ m$^2$ and, hence, closer to the planetary sensitivity (0.267 K W$^{-1}$ m$^2$ for $[dT_s/dT_p] = 1$) if based on skin surface temperature without surface mixing. Without mixing (black circles), the value of $\lambda_a$





decreases strongly with height, because heating at higher levels is less efficient in radiative surface warming. With strong diffusive mixing (red symbols), $\lambda_a$ approaches a constant because the heating is distributed quickly over the troposphere regardless of the layer height. The value of $\lambda_s$ is close to a constant because $RF_s$ already accounts for the fast temperature profile adjustment. Therefore, $RF_s$ is a better measure for surface temperature change than $RF_a$.

The response to ghost forcing characterizes the thermal response for a fixed atmosphere. In addition to mixing, the thermal response depends, of course, on the temperature and composition of the atmosphere. Large changes result from added clouds or from changes in air composition such as humidity. Figure 10a (cyan symbols) shows that ghost forcing below the cloud causes a larger surface temperature change when the reference atmosphere is covered with 100 % cirrus of visible optical thickness $\tau = 3$ at 10 to 11 km altitude. The cloud reduces the heat loss to space. The cirrus cloud must be quite thick to
effectively shield the lower troposphere from radiative heat losses. Note that the infrared absorption optical thickness is typically only half of the visible optical thickness (Garnier et al., 2012). Hence, even for 100 % cover, the solar optical thickness must exceed about 2 to cause a notable reduction on radiative heat losses from the troposphere to space. The plot also shows that increasing the humidity profile to 150 % of the initial value uniformly at all altitudes in the reference atmosphere reduces surface warming by ghost forcing slightly. A uniformly higher humidity in the atmosphere enhances the
infrared layer emissivity, causing stronger local cooling from a ghost layer to space; it also increases the optical thickness between the layer and the surface, reducing surface temperature changes. This is no contradiction to the fact that increases in stratospheric water vapor (and $CO_2$) act to cool the stratosphere but to warm the troposphere (Shine and Sinha, 1991; Solomon et al., 2010). We applied the code also for the tropical standard atmosphere (Anderson et al., 1986). In the more humid tropics with higher and colder tropopause, the relaxation time scales are about 20 % smaller than at mid-latitudes. For
an atmosphere with doubled $CO_2$, the changes are qualitatively similar to increased $H_2O$, but of smaller magnitude. High and thick clouds are far more efficient in changing the radiative relaxation time scales in the troposphere than added $H_2O$ or $CO_2$.

**3.2 Cirrus in comparison to $CO_2$**

In this section we consider the temperature changes induced by a cirrus example, a thin homogenous cirrus layer at 10 to 11 km altitude, with 3 % coverage in an otherwise fixed Earth-atmosphere system. The cirrus ice water content is adjusted to an
optical thickness $\tau = 0.3$ at 550 nm wavelength, and the effective radius of the hexagonal ice particles in this model is set to 20 µm, typical for aged contrail cirrus (Minnis et al., 2013). The net instantaneous RF is positive for the LW and "normal" (SW+LW) cirrus cases and negative for SW cirrus (see Table 1). For comparison, we also consider a 10 % increase in $CO_2$ (360 to 396 µmol mol$^{-1}$) again for an otherwise fixed climate system. Figure 7 and Figure 8 show the instantaneous radiative flux changes and heating rates for added SW, LW and normal cirrus and for increased $CO_2$. Among others, the heating rate
profile for cirrus depends strongly on the assumed optical thickness of the cirrus. For thicker cirrus, the LW heating increases on average over the cirrus but may get negative at top of the cirrus (Liou, 1986). The large heating rate in the air layer at the fixed-temperature surface reflects the finite net downward radiative fluxes at that surface.





For cirrus, we see strongly different temperature responses for the SW and LW cirrus, at least for weak turbulent mixing (Figure 11). The SW cirrus causes a slight warming inside the cirrus by solar radiation absorption (Stackhouse and Stephens, 1991). The main effect of the SW cirrus is a cooling of the lower troposphere culminating at the Earth surface. The LW cirrus enhances infrared absorption inside the cirrus and slightly warms the troposphere below the cirrus by emission from

the cirrus. In addition, LW cirrus enhances the radiation budget at the Earth surface causing a slight warming, but the SW cooling dominates. Only for strong vertical mixing, the heat induced by the cirrus in the upper troposphere gets transported downwards quick enough compared to radiative losses to effectively warm the surface. Convective mixing is weak in this example because the cirrus stabilizes the atmosphere below the cirrus. Convective mixing occurs again only in the uppermost troposphere, between the cirrus layer and the TP.

We note that the cirrus also cools the surface in a case with $Q_0 = 0$, i.e. without fixed dynamical heating, for otherwise the same parameters (most important are albedo and SZA), see Figure 5 and Figure 6. In radiative equilibrium without mixing, again, the cirrus warms the tropopause region but cools the lower troposphere and the surface because of dominant SW changes. The given cirrus cools strongest without mixing but cools also with convective adjustment because the cirrus stabilizes the mid troposphere. Only in case of strong and vertically uniform mixing, positive RF causes a positive

temperature change throughout the troposphere and at the surface.

The $CO_2$ case shows tropospheric warming as expected (Ramanathan and Coakley, 1978; Manabe and Stouffer, 1980; Ogura et al., 2014). The initial heating, mainly from LW radiation, is positive but small ($<0.022$ K $d^{-1}$) in the troposphere and negative in the upper stratosphere with far larger magnitude ($-0.6$ K $d^{-1}$ at 60 km). The literature shows a range of results for $CO_2$ induced heating rates (Collins et al., 2006; Dietmüller et al., 2016). Enhanced $CO_2$ not only heats the troposphere, it also

increases the downwelling LW flux reaching the surface. Convective adjustment occurs for this atmosphere only in the middle and in the upper troposphere; the other parts remain stably stratified. The larger global mean upper tropospheric temperature response in climate models (Hansen et al., 1997a) results from amplification by various climate system changes not included in this model. At high latitudes, reduced vertical mixing, besides sea ice albedo changes, would enhance LW warming at the surface from increased $CO_2$ (Wetherald and Manabe, 1975).

Table 1 lists the computed values for $RF_i$ (at TP, TOA, and surface), $RF_a$ and $RF_s$ at the TP, $\Delta T_s$, and related $\lambda_a$, $\lambda_s$ and efficacy values $e_a$, $e_s$, with respect to $CO_2$, without and with diffusive mixing. The results for convective mixing are close to those without mixing and not shown, therefore. The instantaneous and stratospheric adjusted values apply to fixed troposphere and are, hence, independent of tropospheric mixing.

For $CO_2$, $RF_i$ is positive throughout the atmosphere. $RF_a$ at the TP is in between the $RF_i$ values at TOA and at the TP,
consistent with earlier results (Stuber et al., 2001; Dietmüller et al., 2016). The effective $RF_s$ for fixed climate system is in between the $RF_i$ values at the TP and at the surface.

For cirrus, Table 1 shows that $RF_a$ is small and not much different from $RF_i$, consistent with Dietmüller et al. (2016). The $RF_s$ values for cirrus differ strongly from $RF_a$, even with different sign in the case without diffusive mixing. For SW and





LW cirrus separately, the ratio $RF_s/RF_i$ increases strongly with vertical mixing, e.g., from 0.22 to 0.90 for LW cirrus. At steady state, more and more of the heat induced by the cirrus reaches the surface and less leaves to space for increased mixing. Surface heating (or cooling) is more efficient in heating the underlying compartment (larger $RF_s/RF_i$) than upper tropospheric heating. For the LW+SW cirrus, the SW and LW results for RF and temperature add linearly. However, the

sensitivities and efficacies change nonlinearly because they are ratios of RF and $\Delta T_s$ values. Based on $RF_a$, the efficacy of SW cirrus is larger than for LW cirrus. Hence, efficacies derived from stratosphere-adjusted RF depend on the heating profiles and the mixing. Based on $RF_s$, the efficacies for the well-defined cases are close to unity. They are all close to one, because the cirrus and $CO_2$ changes are small disturbances of the same climate system and the modelled climate systems remain similar also after fast adjustments in all these cases.

Though the nature of the ghost forcing is different, the insight gained in the previous section, consistent with Hansen et al. (1997a), helps to understand the temperature changes induced by cirrus. For weak mixing, $\Delta T_s$ is highly sensitive to the altitude in which the cirrus heating is induced. Also the dependence of $\lambda$ on mixing and the usefulness of effective $RF_s$ to estimate $\Delta T_s$ with nearly constant $\lambda_s$, apply similarly for cirrus. Similar efficacies can be expected only for similar atmospheres and strong mixing. A thick added cirrus changes the atmosphere strongly and causes not only additional

warming but also reduces heat loss from the surface and from the atmosphere below the cirrus to space. In all cases, we find that the effective $RF_s$ is in between the values of $RF_i$ at the TP and at the surface. This finding may be helpful for estimating $RF_s$ for given instantaneous RF.

In the contrail climate study with a global circulation model by Ponater et al. (2006) a plot of the zonal mean vertical cross-section of annual mean temperature response in the equilibrium climate shows that the contrail-induced warming is a

maximum in the upper troposphere and limited to the latitude band in which contrails formed. Hence, the mixing was not strong enough to disperse the contrail-induced warming uniformly over the troposphere. The different efficacies found by Rap et al. (2010a) and by Ponater et al. (2006) may be caused by different ratios of SW to LW RF magnitudes and different vertical mixing in the different models, besides different feedbacks.

Figure 12 illustrates the altitude, scale and mixing dependent timescales of temperature relaxation inside the atmosphere.

Here we show temperature profiles as a function of time starting from steady state for the given cirrus and given mixing model over an adiabatic land surface, after the cirrus is suddenly taken away. As expected from the ghost forcing results, the temperature change returns to zero most rapid at the surface (reaching half its initial value within one time step, 0.25 d); the temperature within the cirrus layer also returns to zero quickly (6.5 d) because of the relatively small geometrical cirrus depth, while the thicker troposphere needs 22.5 d to reach half its initial value. For constant surface temperature, the

relaxation times would be smaller. Convective mixing does not change the results much for this atmosphere. The diffusive mixing reduces both the temperature maximum and the mixing relaxation times scales for local temperature disturbances considerably. Of course, thermal inertia of an ocean would increase heat residence times to many years (Hansen et al., 1985).





### 4 Implications and discussions on regional effects and feedbacks

The results have obvious implications. If we assume forcing by a regional cirrus change and advection by horizontal wind, then any surface cooling or warming will be limited regionally to the immediate neighborhood of the domain with cirrus changes while the upper troposphere warming may travel over large distances. The radiative forcing by cirrus contributes to

long-term global warming only when the heat captured by the cirrus reaches the ocean. A globally uniform heating from localized forcing is unlikely unless advection and mixing occur at timescales far shorter than radiative relaxation. Advection of heat from cirrus or contrail warming has been noted in previous simulations (Ponater et al., 1996; Rind et al., 2000), but the role of radiative cooling has not yet been discussed. Spatial variability in the forcing/response relationship has been derived from climate models for aerosol forcing (Shindell et al., 2010). Hence, efficacy differences are to be expected on

where over continents and oceans the cirrus formed. For small mixing and radiation relaxation time scales also the time scales of the disturbances itself (e.g., minutes to days for contrails and cirrus) influence the mean efficacy of the related RF, because local warming radiates more quickly to space than well mixed warming.

The results presented so far were obtained including fast temperature changes and mixing for otherwise fixed atmosphere, without taking other changes of the climate-system (feedbacks) into account. As a consequence of temperature

change, the climate system will change in many respect (Stephens, 2005). Here we add some discussion to this. Because of different heating profiles and incomplete mixing, temperature change profiles are different and, hence, feedbacks for cirrus will be different from those for $CO_2$.

For illustration, we apply our model also with absolute humidity adapted to temperature changes for fixed relative humidity (Manabe and Wetherald, 1967). For cirrus, because of local warming, such a change enhances humidity mainly in

the cirrus itself. Water vapor is a particularly efficient greenhouse gas near the TP, and added water vapor increases the surface temperature (Shine and Sinha, 1991), consistent with our results, see Figure 13. $RF_s$ is computed for the atmosphere with fixed humidity assuming that the change in humidity (e.g., because of ocean warming) is a slow process. Table 2 lists the temperature changes, climate sensitivities and efficacies in steady state with and without humidity feedback and a feedback factor F, i.e., the ratio of $\Delta T_s$ with and without humidity changes.

The efficacies and feedback factors for cirrus with LW warming and SW cooling heating rates are highly sensitive to small system changes ("ill-conditioned") because the RF is the difference of two large contributions and the sign of $\Delta T_s$ and RF may differ when both are close to zero. We see that the efficacies and feedback factors for SW and LW cirrus differ from one. In contrast to efficacies for $RF_a$, the efficacies for $RF_s$ in the atmosphere with humidity changes are larger for LW cirrus than for SW cirrus. Both are different from one. Hence, neither $RF_a$ nor $RF_s$ are direct measures of the equilibrium surface

temperature change. In the cirrus case, LW forcing gets enhanced while SW forcing gets reduced by climate system changes from changed humidity. Kashimura et al. (2017) investigate surface cooling by added stratospheric aerosol and also find reduced SW RF by reduced humidity and low-level clouds. Ultimately, the role of climate system changes for the RF cannot be determined with a simple model. It requires simulations with a comprehensive climate model.





## 5 Conclusions

Surface temperature changes induced by radiative disturbances depend on the vertical distribution of the radiative heating induced by the disturbances in the troposphere. Since cirrus introduces warming and cooling contributions at different altitudes, the surface temperature response to radiative forcing by added cirrus and contrails is particularly sensitive to the

vertical heating rate profile. It requires strong vertical heat transport by mixing to distribute the induced heat uniformly over the whole troposphere. The mixing has to act at time scales quicker than the radiative heat transfer to avoid loss of energy by radiation to space before the heat can reach the surface. Cirrus tends to stabilize the atmosphere with reduced convective mixing, enhancing the sensitivity to the vertical distribution of the radiative heating.

This paper discussed the relationship between radiative forcings and surface temperature changes in a qualitative

manner based on a radiative-convective-diffusive model. Various RF versions are considered, including instantaneous, stratosphere-adjusted, and effective RF, i.e., $RF_i$, $RF_a$, and $RF_s$. Here, $RF_s$ is computed for fixed surface temperature and the limited set of adjustments represented in the model. After adjustment by thermal relaxation, the $RF_s$ was found to be in between the $RF_i$ values at TOA and at the surface and smaller in magnitude than the corresponding $RF_a$ values. As an extreme, for weak tropospheric mixing, added cirrus may cool the surface even when $RF_i$ and $RF_a$ suggest warming. In

agreement with earlier studies, we find that the climate sensitivity to $RF_a$ varies strongly between the various forcing types while the sensitivity to $RF_s$ is closer to constant. However, when the climate system changes beyond what is included in the fast adjustments considered for $RF_s$, e.g., by humidity changes during ocean warming, the efficacies vary between the forcing types also for $RF_s$. For cirrus including LW and SW effects, no simple relationship between net radiative forcing and temperature change exists.

The radiative relaxation time scales of the disturbance-induced temperature profile changes are of order hours near the surface to months in the mid stratosphere. Hence, temperature changes induced by cirrus near the surface are short-lasting and may be more regionally limited, while upper tropospheric temperature changes last longer and may spread over a larger part of the Earth.

The classical RF concept assumes sufficiently strong mixing within the troposphere, i.e., mixing time scales shorter

than the time scales of thermal relaxation by radiation. One climate model study (Ponater et al., 2005) indicates that the mixing of contrail-induced warming is too weak to mix the heat over the troposphere uniformly. Hence, the contrail warming is distributed over a smaller domain and lasts shorter than for $CO_2$ and this, besides different feedbacks, may cause different efficacies.

These findings may have implications for the assessment of the climate impact of aviation by contrail cirrus. So far,

equilibrium warming from contrails is computed using estimates of RF ($RF_i$ or $RF_a$) together with $CO_2$ climate sensitivity corrected by a contrail efficacy (Ponater et al., 2006; Lee et al., 2009a; Frömming et al., 2012). The net RF for cirrus is often far smaller than the magnitude of its SW and LW parts. In this study we found that the efficacies for SW and LW parts may differ. Hence, the efficacy-weighted RF may be much different from previous estimates.





This study adds further insight into why the RF model is not a universally applicable method to estimate and compare the climate change contributions from various disturbances. A suggestion for an alternative to the RF concept, based on a temperature forcing concept, will be described in a follow-on paper to this study.

5 **Acknowlegments.** Stimulating discussions with Klaus Gierens, Michael Ponater, and Robert Sausen are gratefully acknowledged.

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



**Table 1.** Radiative Forcing (RF) for cirrus and $CO_2$ for fixed climate system (i: instantaneous at tropopause (TP), top of atmosphere (TOA), and surface (SUR); a: adjusted at TP; s: effective at TP), equilibrium air surface temperature changes $\Delta T_s$, (assuming instantaneous heat mixing between surface and lowest model layer) and sensitivity parameters $\lambda$ and efficacies e relative to adjusted and effective $RF_a$ and $RF_s$. The first four rows are the radiative cases with zero turbulent fluxes, the last four rows apply for strongly diffusive cases. The instantaneous and adjusted RF values are the same for both mixing versions. Negative $\lambda$ and e values for cirrus are considered ill-conditioned because highly sensitive to small changes in forcing and mixing contributions.

| | $RF_i$ | $RF_{i,TOA}$ | $RF_{i,SUR}$ | $RF_a$ | $RF_s$ | $\Delta T_s$ | $\lambda_a$ | $\lambda_s$ | $e_a$ | $e_s$ | $RF_s/RF_{i,TOA}$ |
|---|---|---|---|---|---|---|---|---|---|---|---|
| | W m$^{-2}$ | W m$^{-2}$ | W m$^{-2}$ | W m$^{-2}$ | W m$^{-2}$ | K | K W$^{-1}$ m$^2$ | K W$^{-1}$ m$^2$ | 1 | 1 | 1 |
| | | | | | | radiative | | | | | |
| $CO_2$ | 0.83 | 0.41 | 0.07 | 0.72 | 0.27 | 0.12 | 0.17 | 0.45 | 1.00 | 1.00 | 0.37 |
| SW Cirrus | -0.81 | -0.80 | -0.56 | -0.81 | -0.63 | -0.28 | 0.35 | 0.45 | 2.12 | 0.99 | 0.79 |
| LW Cirrus | 0.92 | 0.88 | 0.09 | 0.90 | 0.20 | 0.09 | 0.10 | 0.45 | 0.60 | 0.99 | 0.22 |
| Cirrus | 0.11 | 0.08 | -0.47 | 0.10 | -0.43 | -0.19 | -2.00 | 0.45 | -12.09 | 0.99 | -4.49 |
| | | | | | | radiative and diffusive | | | | | |
| $CO_2$ | 0.83 | 0.41 | 0.07 | 0.72 | 0.70 | 0.19 | 0.26 | 0.41 | 1.00 | 1.00 | 0.97 |
| SW Cirrus | -0.81 | -0.80 | -0.56 | -0.81 | -0.80 | -0.21 | 0.26 | 0.30 | 1.02 | 1.00 | 0.99 |
| LW Cirrus | 0.92 | 0.88 | 0.09 | 0.90 | 0.81 | 0.21 | 0.24 | 0.40 | 0.92 | 1.00 | 0.90 |
| Cirrus | 0.11 | 0.08 | -0.47 | 0.10 | 0.01 | 0.00 | 0.04 | -0.02 | 0.15 | 1.00 | 0.15 |



**Table 2.** $RF_s$ in the atmosphere with fixed humidity and temperature changes $\Delta T_s$ without and with humidity feedback (first 4 and last 5 columns), for radiative and for radiative-diffusive equilibrium (first and last 4 rows). For both feedback variants, the table lists: $\Delta T_s$, $\lambda_s$ and $e_s$ (symbols as in Table 1); the last column is the feedback factor F, i.e., the ratio of $\Delta T_s$ with and without humidity changes. The efficacies and feedback factors for cirrus including LW and SW effects are again considered ill-conditioned.

| | fixed H$_2$O | | | | fixed RH | | | |
| --- | --- | --- | --- | --- | --- | --- | --- | --- |
| | $RF_s$ | $\Delta T_s$ | $\lambda_s$ | $e_s$ | $\Delta T_s$ | $\lambda_s$ | $e_s$ | F |
| | W m$^{-2}$ | K | K W$^{-1}$ m$^2$ | | K | K W$^{-1}$ m$^2$ | 1 | 1 |
| radiative | | | | | | | | |
| CO$_2$ | 0.27 | 0.12 | 0.45 | 1.00 | 0.45 | 1.71 | 1.00 | 3.80 |
| SW Cirrus | -0.63 | -0.28 | 0.45 | 0.99 | -0.57 | 0.90 | 0.52 | 2.02 |
| LW Cirrus | 0.20 | 0.09 | 0.45 | 0.99 | 0.53 | 2.66 | 1.55 | 5.95 |
| Cirrus | -0.43 | -0.19 | 0.45 | 0.99 | -0.05 | 0.11 | 0.06 | 0.24 |
| radiative-diffusive | | | | | | | | |
| CO$_2$ | 0.70 | 0.19 | 0.26 | 1.00 | 0.34 | 0.49 | 1.00 | 1.85 |
| SW Cirrus | -0.80 | -0.21 | 0.26 | 1.00 | -0.41 | 0.51 | 1.04 | 1.93 |
| LW Cirrus | 0.81 | 0.21 | 0.26 | 1.00 | 0.45 | 0.55 | 1.13 | 2.10 |
| Cirrus | 0.01 | 0.00 | 0.24 | 0.92 | 0.04 | 2.61 | 5.35 | 10.70 |




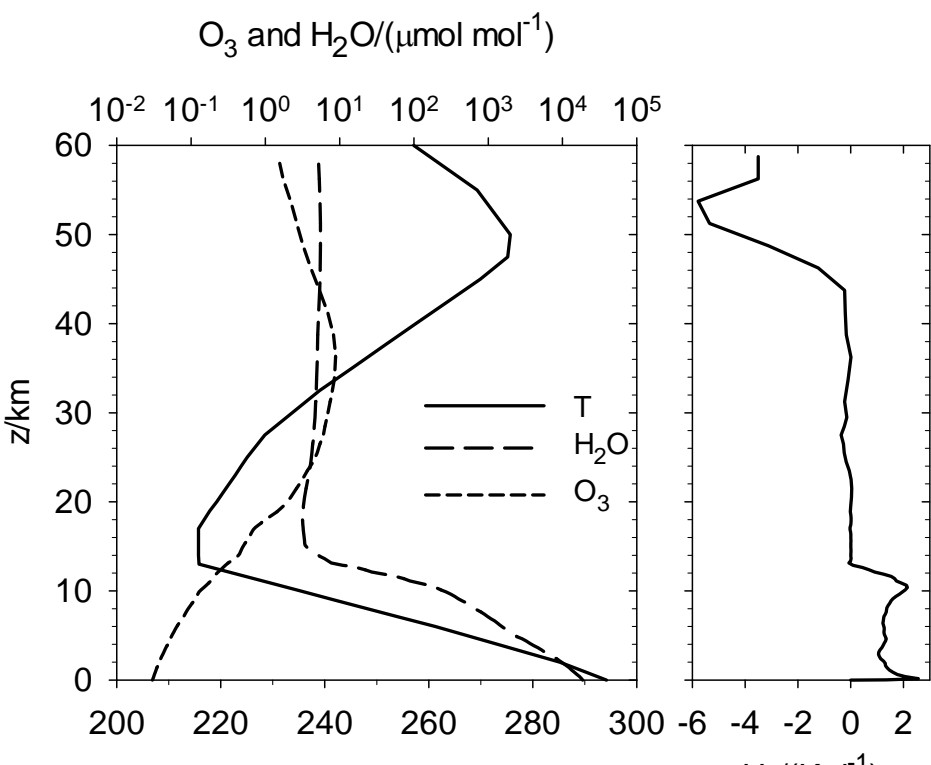

**Figure 1.** Temperature T of the mid-latitude summer standard atmosphere versus height z, together with water vapor and ozone molar mixing ratio ($O_2$: 0.2002 mol mol$^{-1}$; $CO_2$: 360 μmol mol$^{-1}$), and heating rate $H_0 = Q_0/(\rho\ c_p)$ keeping the atmosphere at steady-state, for fixed surface temperature, albedo 0.3, and cos(SZA)=0.25. In the mass-weighted average, $H_0$ = 1.39 K d$^{-1}$ in the troposphere and -0.062 K d$^{-1}$ in the stratosphere.



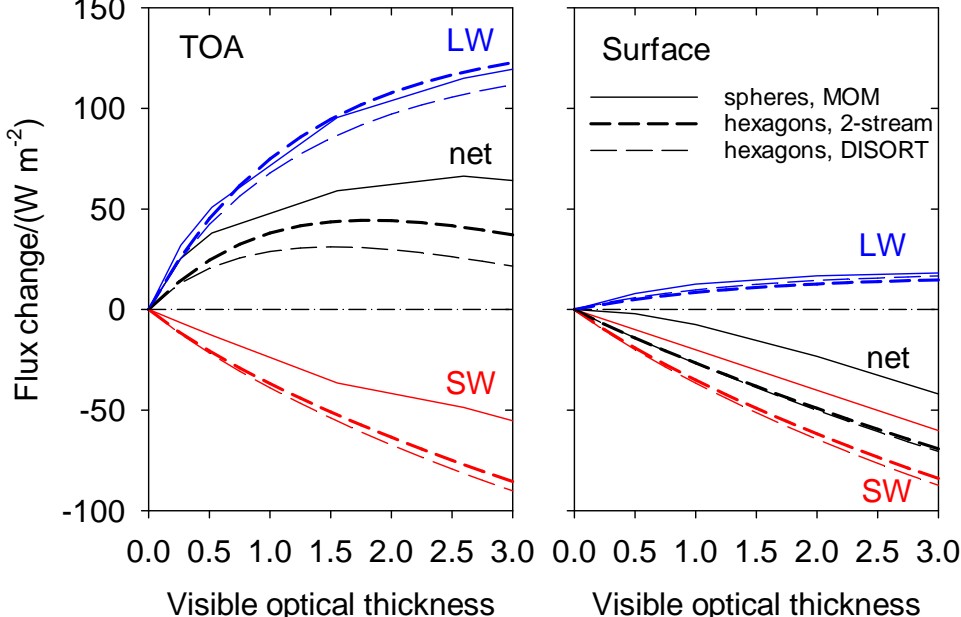

**Figure 2** Day-mean flux changes versus 550-nm optical thickness τ for a homogeneous cirrus layer at 10 to 11 km altitude composed of spheres (Meerkötter et al., 1999) or hexagons (Fu and Liou, 1993), computed with matrix operator method (MOM; (Plass et al., 1973)), two-stream, and discrete ordinate (DISORT) solvers and the Fu & Liou parametrization for molecular absorption for daily mean at 45°N, 21 June, standard mid-latitude summer atmosphere over a surface with albedo 0.2 and fixed surface temperature equal to the surface atmosphere temperature (294.2 K). Differences between the fluxes for these two solvers are of order 10 to 20 %, but DISORT takes orders of magnitude more computing time.





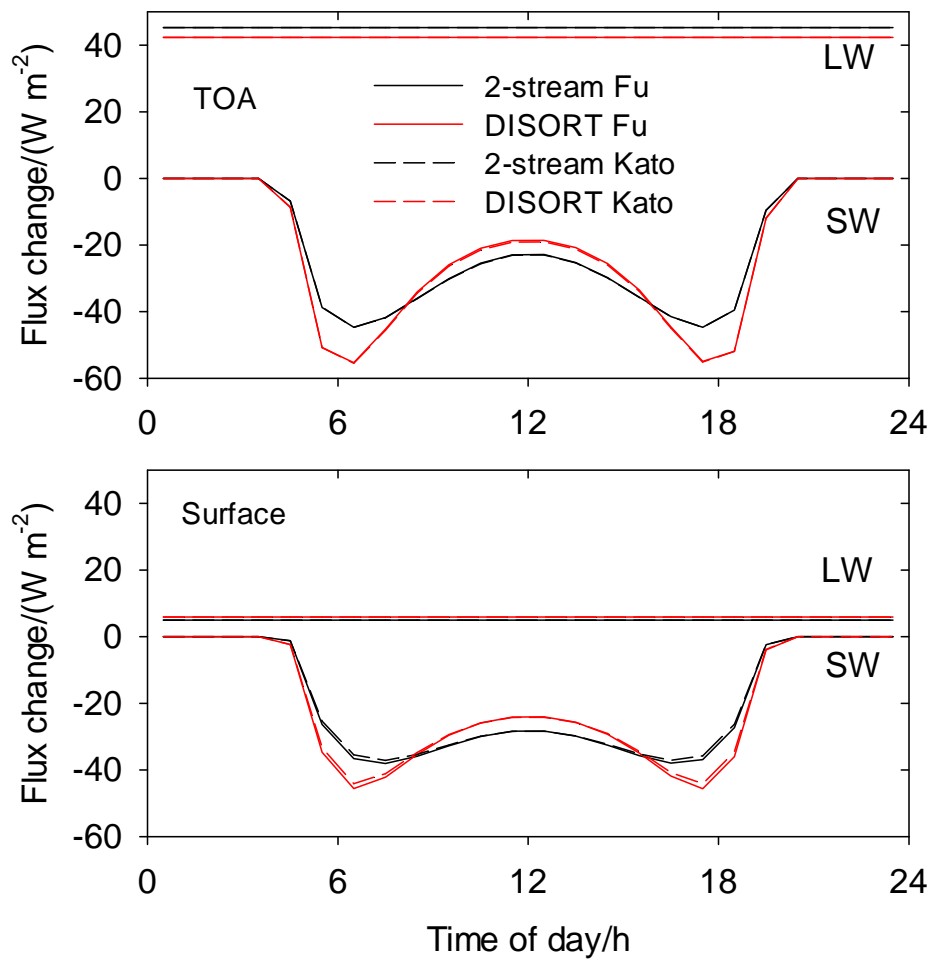

**Figure 3** LW and SW flux changes versus time of day at TOA and at the surface, for two-stream and DISORT solvers, and for Fu & Liou and Kato shortwave molecular absorption parametrizations. The model parameters are the same as in Figure 2, for $\tau$ =0.5. The flux differences for different molecular absorption models of Fu and Kato are far smaller than between the two-stream solver and DISORT.





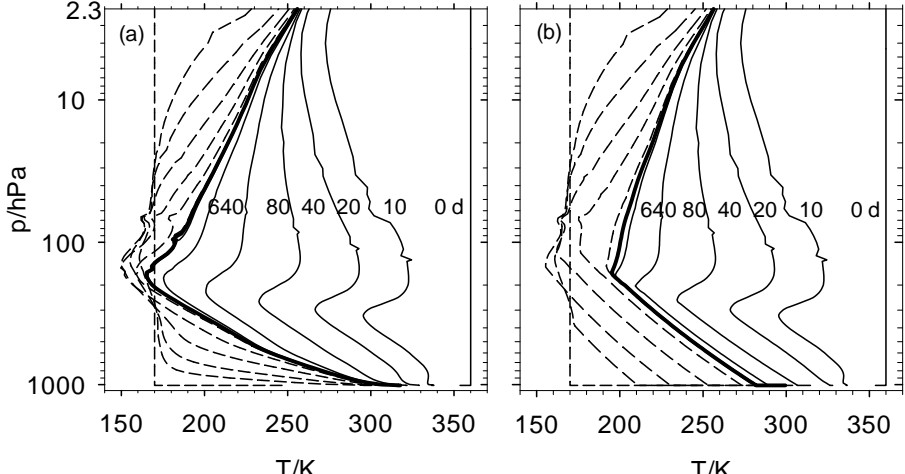

**Figure 4.** Temperature profiles versus pressure altitude (about 0 to 40 km height) starting from 170 K (dashed) and 360 K (full curves) initially, for comparison with Manabe and Strickler (1964), showing the approach to radiative equilibrium, (a) for pure radiative equilibrium and (b) with convective mixing. The model is applied for the cloud-free and aerosol-free mid-latitude-summer-atmosphere composition, with tropospheric $CO_2$ mixing ratio set to 360 μmol mol$^{-1}$, cos(SZA) = 0.25, Lambertian surface with albedo = 0.3 and emissivity= 1. Curves are shown for times 0, 10, 20, 40, ..., 640 d as partially identified by labels. The thick curves show the temperatures after 640 d. The final temperatures from the two initial conditions differ by less than 10$^{-3}$ K in the troposphere and by 0.2 K near 100 hPa.



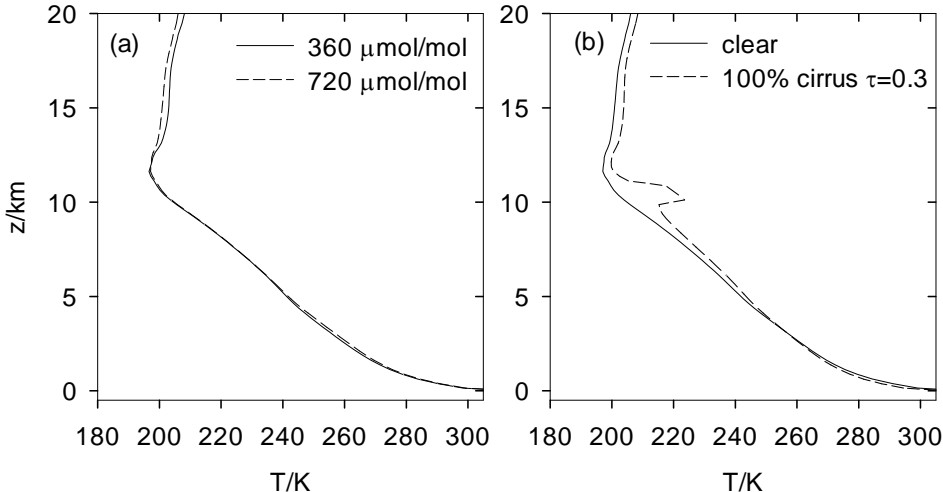

**Figure 5**. Pure radiative equilibrium temperature profiles versus height (a) for reference and for doubled $CO_2$ mixing ratio.
(b) Same for reference atmosphere and atmosphere with a 100-% coverage by a cirrus layer at 10-11 km height with 550-nm
optical thickness of 0.3. The doubled $CO_2$ causes strong stratospheric cooling and a weak tropospheric warming. The cirrus
causes a warming in the stratosphere and upper troposphere but a cooling in the lower troposphere and at the surface.





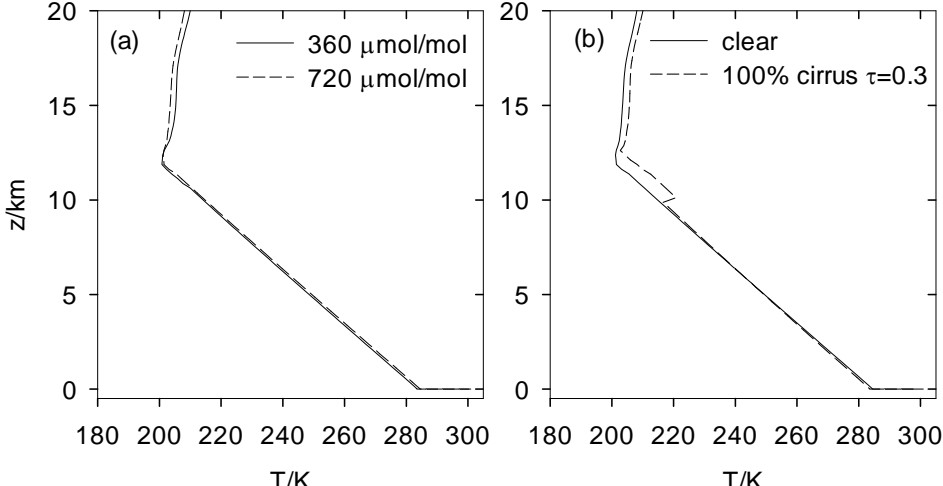

**Figure 6**. Same as Figure 5 with convective mixing. The warming/cooling effects have still the same signs. Convection causes heat exchange leading to warming in the mid-troposphere. With convection, a temperature inversion forms below the given cirrus layer.





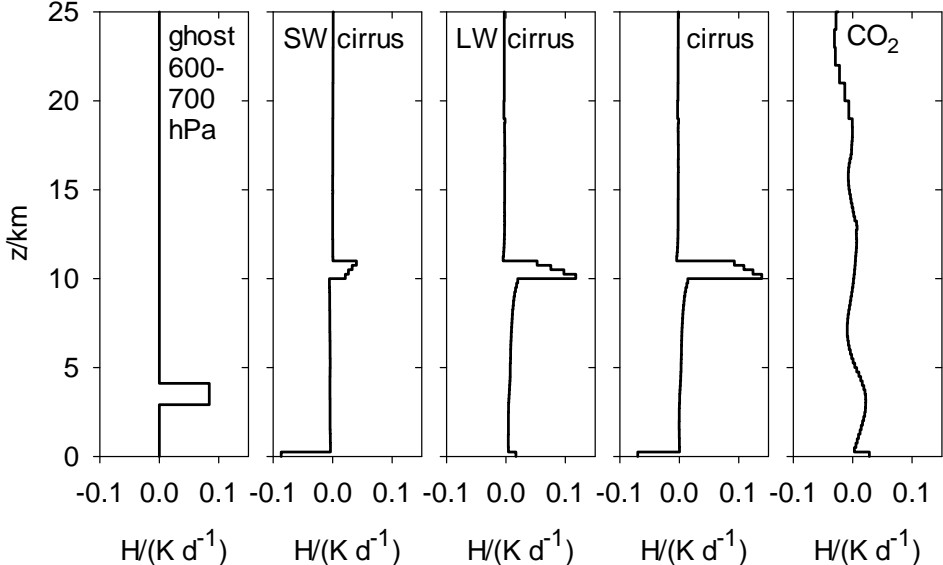

5  **Figure 7:** Initial radiative heating rates H(t=0, z) versus height z for a ghost forcing example, for SW cirrus, LW cirrus, normal cirrus, and for a $CO_2$ disturbance. For plotting, the local heating rate induced by the nonzero radiative fluxes at the fixed-temperature surface is distributed over the lowest 275 m height (same heat capacity as 1 km thick cirrus layer at lower pressure).





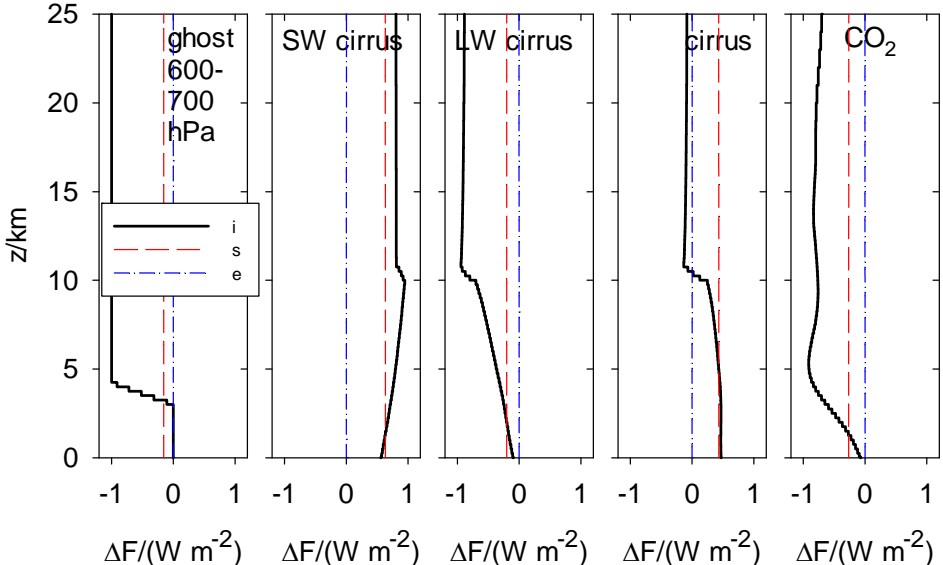

**Figure 8.** Initial (instantaneous) and final (stratosphere-adjusted or equilibrium) net radiative flux changes $\Delta F$ versus height $z$ as induced by a disturbance from added ghost heating, SW cirrus, LW cirrus, "normal" cirrus with SW and LW contributions, and 10 % increased $CO_2$, in the panels from left to right, respectively. Black full lines: instantaneous flux; red dashed line: adjusted to constant surface temperature; blue dash-dotted line: equilibrium over adiabatic surface.




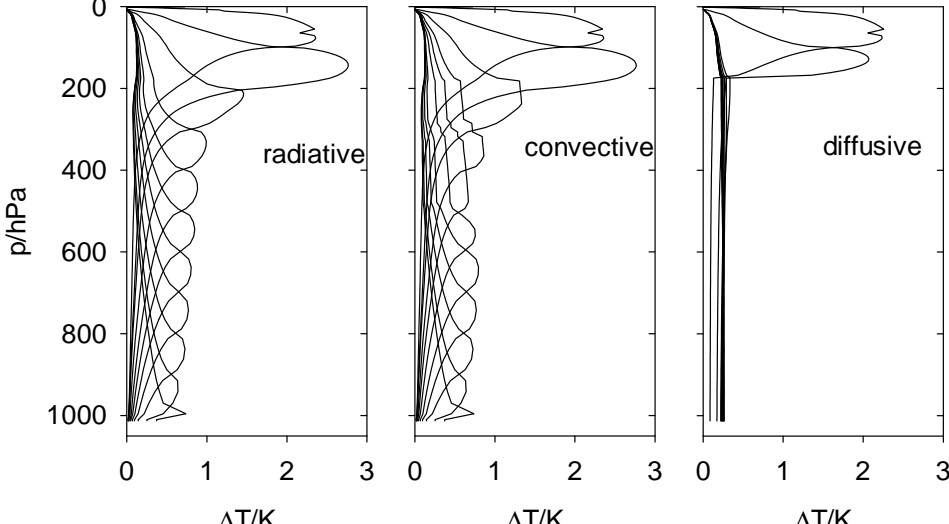

**Figure 9:** Temperature response profiles versus pressure altitude for layer heating (ghost forcing) with 1 W m$^{-2}$ in ten subsequent 100-hPa pressure layers and at the surface for adiabatic surface with rapid surface mixing. Left: radiative with zero turbulent fluxes; middle: radiative-convective mixing; right: for a moderately strong diffusive mixing κ = 100 m$^2$ s$^{-1}$ constant throughout the troposphere.





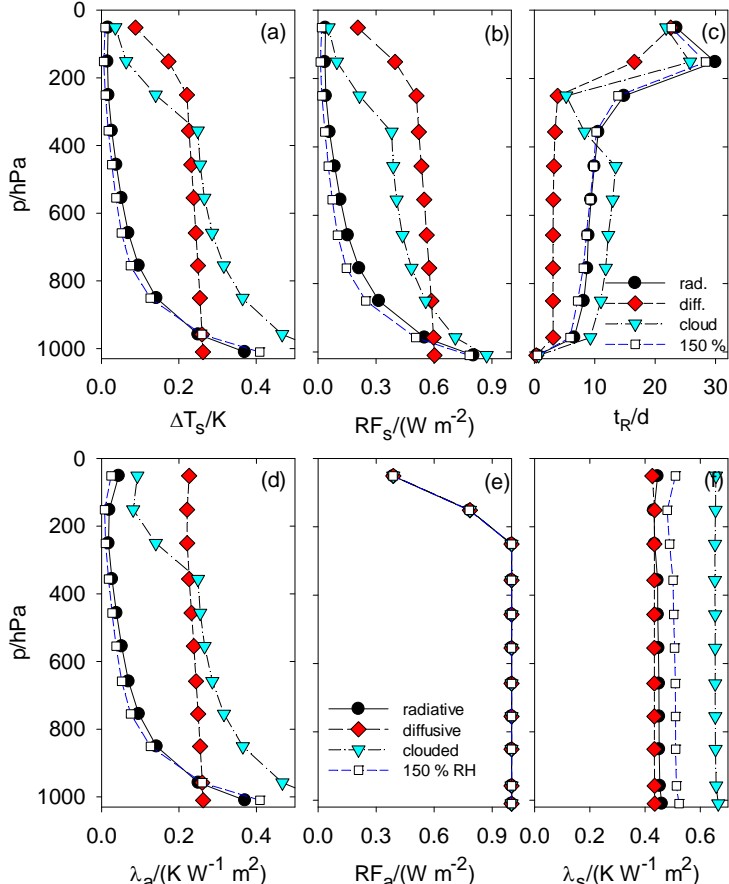

**Figure 10:** (a) Temperature change at the surface for layer heating versus layer pressure height in an atmosphere. The ghost forcing corresponds to an $RF_i$ of 1 W m$^{-2}$ at TOA. Black symbols with full lines: model results for radiative equilibrium without mixing; red diamond: with strong diffusive mixing for $\kappa$=100 m$^2$ s$^{-1}$ in the whole troposphere; cyan triangles: with strong diffusive mixing and a 100 %-coverage cirrus layer with $\tau$ = 3 between 10 and 11 km height; open square with dashed blue line: radiative equilibrium without mixing with 1.5 times enhanced $H_2O$ mixing ratio at all levels in the reference atmosphere. (b) Corresponding $RF_s$ values for fixed $T_s$. (c) Relaxation time scales $t_R = \Delta T_{layer}/H$. (d) Climate sensitivity parameter $\lambda_a = \Delta T_s/RF_a$ based on stratosphere-adjusted $RF_a$; (e) $RF_a$; (f) climate sensitivity parameter $\lambda_s = \Delta T_s/RF_s$ based on effective $RF_s$.





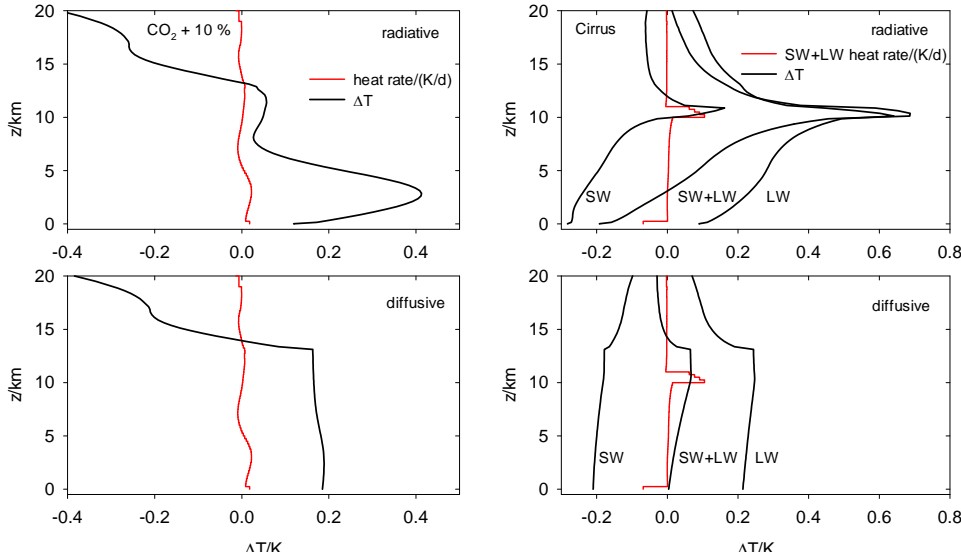

**Figure 11:** Equilibrium temperature change $\Delta T$ in K versus altitude z in km for disturbances by $CO_2$ (left) and by SW, LW
and normal cirrus (right) in an atmosphere above an adiabatic surface with rapid local mixing at the surface (black line), for
radiative equilibrium with zero mixing (top) and with uniform diffusive tropospheric mixing (bottom). The red curves are the
net (LW+SW) initial instantaneous heating ratings in K d$^{-1}$.




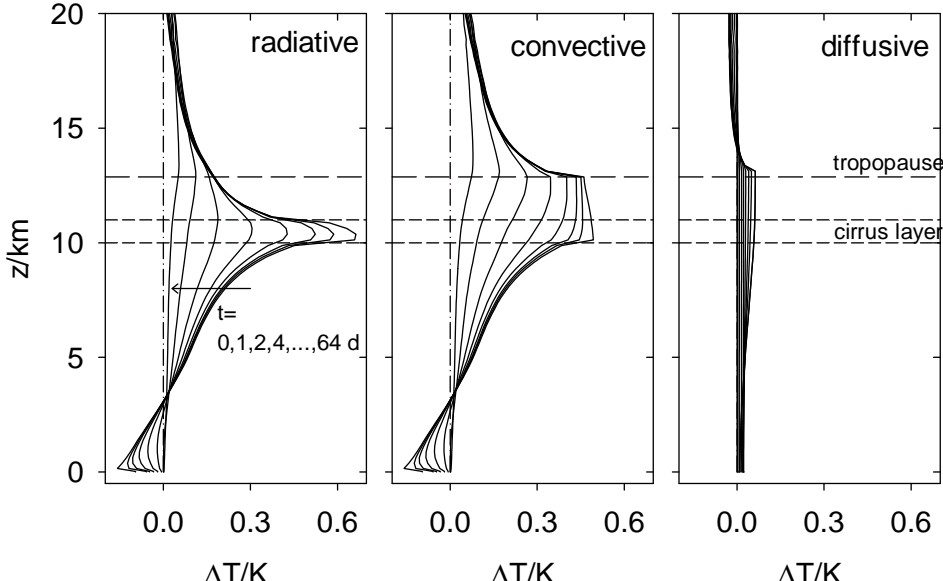

**Figure 12.** Decay of an initial steady-state cirrus-induced temperature increase, at times 0, 1, 2, 4, ..., 64 d after cirrus ceased, for the radiative, radiative-convective and radiative-diffusive mixing cases. Tropopause and cirrus layer heights are indicated by dashed lines. The times needed to reach half the initial values are 0.25 d, 22.5 d and 7 d for the temperature at the surface, on average over the troposphere, and in the cirrus layer, respectively, for the radiative case, and shorter for the other mixing cases.





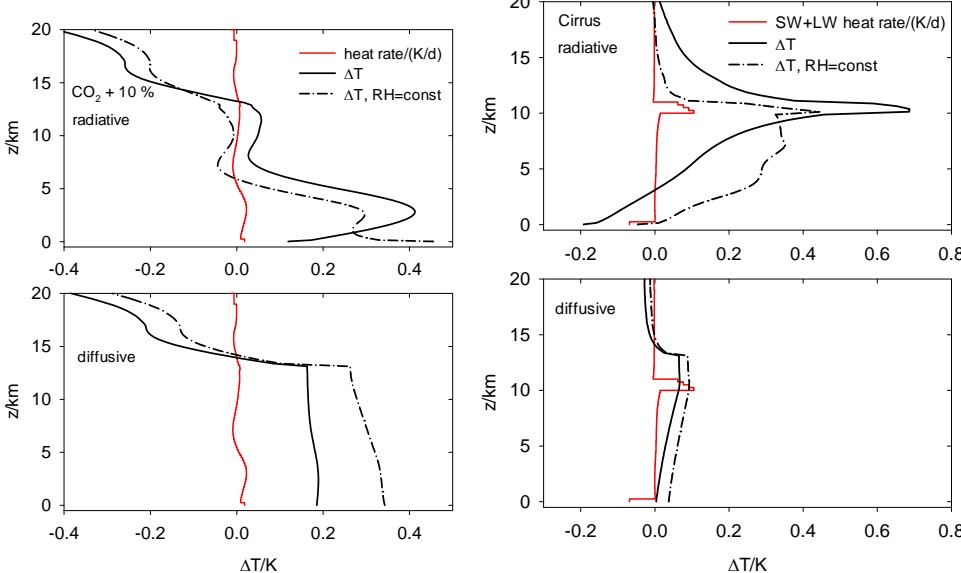

**Figure 13.** As Figure 11, without (full line) and with (dash-dotted) humidity adapted to constant relative humidity RH (left

5 for $CO_2$, right for normal cirrus).