# Peer review of "Sensitivity of surface temperature to radiative forcing by contrail cirrus in a radiative-mixing model"

_Atmospheric Chemistry and Physics, 2017_

## Referee Comment (RC1) · Anonymous Referee #1 · 22 Jun 2017

This paper represents in some ways a rather impressive and stimulating study, but in its present form I am not sure its conclusions are safe ones (in the sense that they do not advance our understanding of real-world climate responses). I feel that the authors may be on to an important point, about the efficiency with which radiative perturbations in the upper troposphere can be transmitted to the surface, but whether the experiments presented here are sufficient to establish that importance is not so clear. I cannot give a strong recommendation for acceptance in anything like its present form. On the other hand I do not wish to discourage the authors from pursuing this important and interesting topic.

[Figure]

One of my issues with the paper is that it oscillates between being a fundamental study of the fate of radiative perturbations in the climate system and being a more applied and directed study concerning contrails in particular, and it is easy for the reader to get lost amongst material that is not clearly relevant. Certainly this reviewer felt lost on several occasions, and I found myself having to go back and re-read earlier sections and still I sometimes struggled. I am sorry to say that if I had not been a reviewer, I may not have persevered with reading the paper.

So, for example, some of the approximations that are made may be appropriate to a more theoretical/illustrative study, are not so clearly appropriate if the aim is to specifically understand contrail efficacy. They might even invalidate the results. And similarly, while it might be useful to discuss the pure radiative equilibrium case in a theoretical study, that case is not really relevant to understanding contrail efficacy. I feel that the repeated presentation of the radiative equilibrium case gets in the way of understanding the real-world response. Overall, I felt the manuscript tried to be too "completist" (e.g. presenting figures and calculations that didn't need to be presented) which made the manuscript longer and more complex than it needed to be.

A central issue in this paper is the ability of real-world cirrus/contrails to distort the vertical profile of temperature in the way that is shown in figures 5 and 6. It is this stabilization that is key to the authors' results. Is there any wide scale evidence that cirrus of contrails do this, particularly away from the rather special conditions in the tropical tropopause layer? I feel the authors need to do a critical analysis of the literature on this point, as the paper would be greatly strengthened if they are able to present any such evidence.

Detailed comments. Those preceded by an asterisk are more major comments.

1:1 I have a concern about the title. I do not believe that cirrus causes radiative forcing. It certainly has a radiative effect that can change (and hence induce a feedback) but this is rather unlike the contrail case. Perhaps "contrails and contrail cirrus" would be

better as these are more obviously forcings.

1:8 "basically without climate system changes" – presumably this means "no feed-backs" except for temperature change?

1:13 "Heat induced by cirrus" – since in principle there is a latent heating associated with cirrus formation, clarify that this is "radiative heating due to cirrus"

1:14 "adjusted" – is this stratosphere-temperature adjustment?

1:23 and throughout: I think it better to talk of a "cloud radiative effect", as is now common in the literature, rather than a "net radiative forcing" of cirrus.

2:2 "heat induced" – maybe better as "changes in radiative heating"

2:7 "covers" –> "is estimated to cover"

2:26 "contrails occur mainly over land" – this could be clearer – do you mean that most flights are over land, or conditions for contrail formation are more likely over land? I think it is the first of these.

*3:8-9 As is discussed by Hansen et al. (and I think in papers by Ponater) it needs to be clear that lambda_co2 is not a fixed number even in a single climate model, as it depends on the size of the CO2 perturbation. Hansen et al are careful to define their efficacy relative to a specific CO2 change (see their para 34 and Table 1), so that other CO2 perturbations have themselves an efficacy that departs from 1 relative to their specific case.

4:1 "rating" – I didn't quite understand this word – perhaps it is "rerouting" afflicted by an automatic spell-checker?

4:18 "similar to a dust layer" – I didn't understand – mineral dust layers (if that is what is meant) can have a LW forcing.

5:9 And F_T is also zero in the radiative equilibrium case, I presume. If so, perhaps the

text should say this.

**5: This page needs much better structure and to establish a consistent terminology. Three cases are presented "pure radiative equilibrium", "radiative-diffusive mixing " and "radiative-convective mixing" . But sometimes different terminology is used. 8:12 refers to the "radiative case" (but all cases are radiative), Fig 10 caption refers to "radiative equilibrium without mixing" and "strong diffusive mixing", Figure 11 refers to "radiative equilibrium with zero mixing" and "uniform diffusive mixing" and then Figure 12 refers to "radiative equilibrium with zero turbulent fluxes" and "moderately strong diffusive mixing". I could go on. I hope the author will see the need to adopt a concise and consistent terminology but also to consider whether a good scientific purpose is served by presenting results for cases in almost all figures. The terminological confusion is further accentuated on page 6 by having two variants to determine the skin temperature – no separate name is given to each case, and I am frankly not sure it is necessary to even present results from both, as the zero surface turbulent heat flux case is entirely theoretical.

*5:15 "model includes a cirrus layer" – I think it is equally important to make clear that it ONLY includes a cirrus layer – i.e. no other cloud layers are included. The paper was not clear on this point but I regard this as a serious restriction when it comes to specifically looking at the impact of contrails, and so it is important that this is kept in mind. The impact of cirrus on the surface LW and SW budget, as well as the radiative heating at cirrus cloud base (e.g. Figure 7), will be considerably affected by the lower level clouds which are missing here.

*5:21 It is not clear where the value for diffusivity comes from. Some earlier study? The value plays such an important role in the analysis that it has to be justified in a more rigorous way. And it is important to again acknowledge important caveats: in this case, vertical heat transport in the real atmosphere is not, for the most part, diffusive, and so what is adopted here is a convenience for the simple model.

5:25-29 It is a little hard to follow this – given the signs shouldn't the "max" on line 26 be a "min"?

*6:10 Using a surface albedo of 0.3 is a very crude way of mimicking low level clouds, and of course only does so in the SW (and so the LW surface budget is more sensitive to atmospheric perturbations than it would otherwise be). It is not quite clear to me why other clouds are excluded – is it an attempt to simplify or a methodological difficulty in including them? And why 0.3? I recognise this is the planetary albedo, but a surface albedo of 0.3 does not yield a planetary albedo of 0.3, because of atmospheric absorption (pushing one way) and Rayleigh scatter (pushing the other). It would be reassuring to know what the control top-of-atmosphere radiation budget is, as this would help determine how realistic the forcings (especially the longwave) are.

*6:10 "cos(SZA)=0.25" – this surprised me too. I understand that this yields the correct incoming solar radiation at top of atmosphere, but the high zenith angle (75 degrees) will significantly bias the SW effect of contrails to be more negative – indeed it is the zenith angle close to the most negative radiative forcing, according to the excellent Schumann et al. (2012 - 10.1175/JAMC-D-11-0242.1) paper and this may significantly affect some of the section 3.2 results . In radiative convective models (such as Manabe and Wetherald) it is common to assume a cos(SZA) of 0.5 and to assume a fractional day length of 0.5, although it may be more preferable to integrate over zenith angle.

*7.10 Following on from the above comment, I am now a bit further confused. In the caption of Figure 2, it refers to the daily mean at 45N on 21 June. How does this relate to the cos(SZA)=0.25? And why is a surface albedo of 0.2 used here when it is 0.3 in the text? I guess Figure 2 is trying to justify the use of the 2-stream hexagons scheme used in the radiative-convective model, but it seems to me that it is not testing it for the conditions applied in that model. I am sorry if I misunderstand. And I have a similar query about Figure 3. Since, from my understanding, the radiative-convective model does not integrate over the diurnal cycle, this plot leaves a somewhat misleading impression and I am not sure of its purpose here. My bigger question is whether the

choice of cos(SZA)=0.25 leads to a bias in the SW budget of contrails. Also since a cirrus optical depth at 500 nm of 0.3 (10:25) is applied in the experiments, it is not clear why a value of 0.5 is used in Figure 3.

7:19 The figures show 360-720 ppb, the text says 300-600 ppb

7:17 Figure 4: While it is useful for the authors to have performed this calculation, I see no reason for including it in the paper – it is a result that is over 50 years old and in my view just inflates the paper. I feel something of the same way about Figures 5 and 6, since they are referred to only in passing. The inversion in Figure 6 may be something of an artefact resulting from the exclusion of lower level clouds

8:6 The expression for heating rate is textbook physics and doesn't need including – I am not sure the value for the lowest level is in any case correct, if the surface pressure is really 1013 hPa (I get 0.64 K/day).

8:12 "radiative EQUILIBRIUM case"

8:13 "smaller vertical scales" – it is hard to see this when the plot is presented in linear pressure.

8:17: I agree that the 8-13 micron window is "more transparent" than neighbouring spectral regions, but it is hardly transparent, because of water vapour continuum absorption in this region.

8:19 "stratosphere" – this sentence only makes sense to me if it is the "lower stratosphere"

8:21 "rather stable" – it is unclear what measure of stability is being used in making such a statement

9:23 Perhaps 2 significant figures are enough in this and later paragraphs?

10:5-20 The experiment described here (100% cirrus, 150% perturbation to humidity) feels very contrived and in my view was a distraction. I suggest it be removed.

*10:24 Why 3% given the 0.2-0.5% at 2:7? But I am concerned that the assumed cirrus amount will ultimately impact the radiative heating in the upper troposphere and hence the extent to which that region can be decoupled from the atmosphere below. In addition, I suspect that the impact is also highly dependent on the height of the cirrus as well as the assumptions about underlying clouds.

11:1 "weak turbulent mixing" – which case is this referring to? See my comment **5. If you mean zero-mixing, the text should say this.

11:6 "only for strong" – but as I understand you have only performed the experiment for zero or strong, so there is no intermediate case? I then get further confused by the discussion of convective mixing later in the paragraph, partly for the reasons discussed above, but partly because it is not shown on Figure 11. I suspect the result is also highly sensitive to the assumed cirrus height. Comparing Figure 5 and 6, it seems clear that convective mixing is impacting the temperature profile throughout the depth of the troposphere so it confused me to say that "convective mixing is weak"

*11:10 The discussion at 6:10 about the chosen solar zenith angle calls into question this result, and I suggest it is revisited.

12:18 I can see no such plot in Ponater et al. (2006) – I am sorry if I miss it. Perhaps the text should refer to Figure 2 of Ponater et al. (2005) (see also 14:25) but even there I am a bit doubtful whether the point being made is the full story; the maximum in upper tropospheric warming may be a result of well-known moist adiabatic processes (in which a surface perturbation is amplified at upper levels via the divergence of moist adiabats with height).

13: I found this discussion rather conjectural and suggest it could be removed

*14: Although the central idea of this paper may indeed prove to be correct, this conclusions need to draw attention to the many caveats about the simple model that is adopted and how these may impact on the final result.

---

## Author Comment (AC1) · 24 Jun 2017

As noted by Referee 1, page 12 - line 18 contains a mistake, which we want to correct: The reference should refer to Figure 2 in Ponater et al. (2005), not Ponater et al. (2006).

---

## Referee Comment (RC2) · Anonymous Referee #2 · 21 Jul 2017

The authors investigate the extent to which top-of-atmosphere forcing from jet contrails are able to influence the surface temperature using a simple radiative-convective-diffusive model. This may be phrased as the "efficacy" associated with such forcing, and the authors show that this efficacy is strongly dependent on the assumed mixing within the model.

The results of this study are of interest, but they are derived from a very simplified model, and because of its simplicity I am a bit unclear on the implications of this study for Earth's atmosphere. In particular:

1) The authors find that in the limit of weak tropospheric vertical mixing, the effect of

upper tropospheric forcing like that of contrails can be to cool the surface. This seems to run counter to GCM studies of Hansen et al. (2006) and Ponater et al. (2006), which show a more constant tropospheric response, presumably because they have some vertical mixing. Does this mean that the weak vertical diffusion case in this study is simply not relevant to Earth's atmosphere?

2) In the mid-latitude case, convection is hardly active because the large-scale forcing $Q_0$ stabilises the atmosphere. But in Earth's atmosphere, convection acts intermittently and the convective mixing is therefore underestimated by this model. Further, I think it is unreasonable to expect $Q_0$ to remain unchanged in response to the forcing. The thermal stratification of the midlatitudes is set by this large-scale forcing, and a change in this thermal stratification will likely have an influence on the midlatitude eddies. Is the vertical diffusion meant to be a parameterisation of these missing processes? If so, what level of vertical diffusion is relevant for Earth's atmosphere?

3) In the "tropical" case ($Q_0 = 0$; Fig. 6) the convective adjustment is controlling the lapse rate, as is the case in Earth's tropics. But here, the forcing applied is very strong: 100% Cirrus cover. In this case, the Cirrus produces an inversion in the upper troposphere, and drives a second convective cell above the tropopause. I'm not sure this is a plausible outcome of contrail forcing. What happens if the forcing is reduced to a cloud cover of 0.2-0.5%? Do you still get the same decoupling from the surface? How does this depend on the height of the forcing?

4) What does Fig 11 look like with radiative and convective adjustment? we expect the CO2 response to warming to be relatively uniform in the troposphere. This is true with high diffusion, but does not seem to be true in the radiative-convective case. To me this suggests that the no diffusion limit is not relevant for the Earth.

My suggestions to improve the manuscript in light of these comments are as follows:

- More consistent forcing levels across the experiments. In some cases the Cirrus cloud cover is set to 3%, in others 100%. Why is this the case? And how was 3%

[Figure]

chosen? It seems much larger than the 0.2-0.5% quoted in the introduction for Contrail fraction. Does the response depend on the size of the forcing? What about the height of the forcing?

- Some more discussion on how the results from the simple model should be interpreted. In particular, what is the level of vertical mixing relevant for Earth's atmosphere in midlatitudes and in the tropics? How does the assumption of diffusive mixing affect the results.

- I think the study would benefit from using single-column model with a more realistic description of convection than the simple model used here (e.g., the single column model of a IPCC-class GCM). While this does not ameliorate all the problems with using a 1-D description of the atmosphere, it will ensure the convective response given the mean state will be somewhat realistic, particularly for the "tropical" case in which $Q_0$ is zero.

Minor comments:

page 1: Line 8: What does "basically without climate system changes" mean? Does this refer to the dynamic heating in the model? This should be clarified here and in the other places where this statement is made in the manuscript.

page 2: line 33: Here it is argued that contrails do not behave the same as high clouds, but later the forcings you apply are described as either thin cirrus or contrails. This contradiction should be resolved

page 3: Line 32: I am not sure what it means to avoid warming contrails. Does this mean that one mitigation option is to move flight paths to regions in which the effects of contrails is a cooling?

page 5: Line 1-10: The discussion here is very confusing. At one point it is stated that $Q_0$ is the sum of the divergence of $F_R$ and $F_T$, but it is a bit unclear whether this statement is supposed to only apply for $T = T_0$ or more generally. Later it is stated

that the Q_0 = 0 case is "pure radiative equilibrium", but I think this should be Q_0 = 0 and F_T = 0.

page 5: line 20: I don't understand why \Gamma drops out of the equation for \Delta T, or why the contribution from \Gamma affects Q_0. Isn't Q_0 fixed? I think the equation for \Delta T should be presented for clarity.

page 6: line 10: Setting the cosine zenith angle to 1/4 biases the solar radiation to have a high zenith angle, this will increase the reflection from clouds and bias the results. For the global mean, one should use the insolation weighted zenith angle (Cronin 2014). But I do not see why the global mean insolation is necessarily desired. The temperature profile used is one of the mid-latitudes, so presumably that is the focus. Why not use a diurnally varying solar insolation for e.g., 45 deg?

page 6: line 20: The radiation only boundary condition for T_skin is unphysical for cases with turbulent fluxes. Perhaps it would make more sense to use an assumed value of the surface enthalpy exchange coefficient and wind speed that are typical of Earth's surface conditions.

page 8: line 32: The Hansen et al. (1997) result needs explaining. What type of model were they using? Does this indicate that the strong mixing limit is the appropriate one?

page 10: line 25: Here 3% Cirrus coverage is used, but the global cover mentioned in the introduction is 0.2-0.5%. Does the magnitude of the Cirrus cover have any effect on the results?

page 11: line 9: It appears that the Cirrus drives convection above it to the tropopause. Is this likely for the forcing from Jet contrails in the next century?

References:

Cronin, T.W. (2014), On the choice of average solar zenith angle, JAS.

Hansen, J., M. Sato, and R. Ruedy: Radiative forcing and climate response, J. Geophys. Res., 102, 6831-6684, 1997.

Ponater, M., S. Brinkop, R. Sausen, and U. Schumann: Simulating the global atmospheric response to aircraft water vapour emissions and contrails. - A first approach using a GCM, Ann. Geophys., 14, 941-960, 1996.

---

## Author Comment (AC2) · 22 Aug 2017

Responses to Reviewer 1.

We thank the reviewer for his thoughtful and detailed review. The review comments lead to considerable changes and several improvements.

This paper represents in some ways a rather impressive and stimulating study, but in its present form I am not sure its conclusions are safe ones (in the sense that they do not advance our understanding of real-world climate responses). I feel that the authors may be on to an important point, about the efficiency with which radiative perturbations in the upper troposphere can be transmitted to the surface, but whether the experiments presented here are sufficient to establish that importance is not so clear. I cannot give a strong recommendation for acceptance in anything like its present form. On the other hand I do not wish to discourage the authors from pursuing this important and interesting topic.

We have revised the paper in several parts. We tried to clarify open issues and to reduce the complexity by deleting any material which we feel is not absolutely necessary to bring over our central point: Mixing is important for climate sensitivity to contrail cirrus.

One of my issues with the paper is that it oscillates between being a fundamental study of the fate of radiative perturbations in the climate system and being a more applied and directed study concerning contrails in particular, and it is easy for the reader to get lost amongst material that is not clearly relevant. Certainly this reviewer felt lost on several occasions, and I found myself having to go back and re-read earlier sections and still I sometimes struggled. I am sorry to say that if I had not been a reviewer, I may not have persevered with reading the paper.

We now revised the paper at several places to reduce material and complexity and hope that the reviewer now finds it more worthwhile to spend time on the text.

So, for example, some of the approximations that are made may be appropriate to a more theoretical/illustrative study, are not so clearly appropriate if the aim is to specifically understand contrail efficacy. They might even invalidate the results. And similarly, while it might be useful to discuss the pure radiative equilibrium case in a theoretical study, that case is not really relevant to understanding contrail efficacy. I feel that the repeated presentation of the radiative equilibrium case gets in the way of understanding the real-world response. Overall, I felt the manuscript tried to be too "completist" (e.g. presenting figures and calculations that didn't need to be presented) which made the manuscript longer and more complex than it needed to be.

We agree in several respect and state now more clearly the merits and limitations of the approach, in Section 3 and in the Conclusions. On the other hand, we found that the results are robust against many model parameters, as now explained in subsections 3.1 and 3.2.

We deleted the discussion on the pure radiative equilibrium case and the related former figures 4 to 6.

A central issue in this paper is the ability of real-world cirrus/contrails to distort the vertical profile of temperature in the way that is shown in figures 5 and 6. It is this stabilization that is key to the authors' results. Is there any wide scale

evidence that cirrus of contrails do this, particularly away from the rather special conditions in the tropical tropopause layer? I feel the authors need to do a critical analysis of the literature on this point, as the paper would be greatly strengthened if they are able to present any such evidence.

The stabilization is not the central issue (a lapse rate change is a classical feedback). The main point is the importance of mixing for surface temperature sensitivity, even for fixed lapse rate. Note, the figures on radiative-convective equilibrium were presented as test cases to demonstrate that the model passes necessary tests. We agree that convective adjustment is not so important for mid-latitude cirrus and reduced these parts considerably, therefore.

Detailed comments. Those preceded by an asterisk are more major comments.

1:1 I have a concern about the title. I do not believe that cirrus causes radiative forcing. It certainly has a radiative effect that can change (and hence induce a feedback) but this is rather unlike the contrail case. Perhaps "contrails and contrail cirrus" would be better as these are more obviously forcings.

We still mean that the study is of relevance beyond contrails, but now decided to change the title and to reduce the scope. So the new title refers more restrictively to contrail cirrus. In the Conclusions we still state: "The findings may apply also for other disturbances."

1:8 "basically without climate system changes" – presumably this means "no feedbacks" except for temperature change?

Yes, and the new wording says this.

1:13 "Heat induced by cirrus" – since in principle there is a latent heating associated with cirrus formation, clarify that this is "radiative heating due to cirrus"

We now talk about "energy induced by radiation".

1:14 "adjusted" – is this stratosphere-temperature adjustment?

Yes, now "adjusted" is replaced by "stratosphere-adjusted".

1:23 and throughout: I think it better to talk of a "cloud radiative effect", as is now common in the literature, rather than a "net radiative forcing" of cirrus.

We now use the term in Figures 1 and 2, e.g. We still feel that the term "net RF" is appropriate for contrail cirrus.

2:2 "heat induced" – maybe better as "changes in radiative heating"

Basically we agree, and write "Heat induced by radiation"

2:7 "covers" –> "is estimated to cover"

We agree, and changed the text accordingly.

2:26 "contrails occur mainly over land" – this could be clearer – do you mean that most flights are over land, or conditions for contrail formation are more likely over land? I think it is the first of these.

We mean that most flights are over land, and clarified this aspect in the text, accordingly.

*3:8-9 As is discussed by Hansen et al. (and I think in papers by Ponater) it needs to be clear that lambda_co2 is not a fixed number even in a single climate model, as it depends on the size of the CO2 perturbation. Hansen et al are careful to define their efficacy relative to a specific CO2 change (see their para 34 and Table 1), so that other CO2 perturbations have themselves an efficacy that departs from 1 relative to their specific case.

We agree that $\lambda_{CO_2}$ depends on the size of the $CO_2$ perturbation, and changed the text into "for a given change in $CO_2$".

4:1 "rating" – I didn't quite understand this word – perhaps it is "rerouting" afflicted by an automatic spell-checker?

We meant "rating" in the sense of "assessing". But now feel that this statement is no longer necessary and deleted this part for shortness and to avoid such misunderstanding.

4:18 "similar to a dust layer" – I didn't understand – mineral dust layers (if that is what is meant) can have a LW forcing.

We tried to find an example that is mainly scattering and in the SW range. We now write: "similar to a layer of small and non-absorbing particles"

5:9 And F_T is also zero in the radiative equilibrium case, I presume. If so, perhaps the C3 text should say this.

We agree and changed the text.

**5: This page needs much better structure and to establish a consistent terminology. Three cases are presented "pure radiative equilibrium", "radiative-diffusive mixing " and "radiative-convective mixing" . But sometimes different terminology is used. 8:12 refers to the "radiative case" (but all cases are radiative), Fig 10 caption refers to "radiative equilibrium without mixing" and "strong diffusive mixing", Figure 11 refers to "radiative equilibrium with zero mixing" and "uniform diffusive mixing" and then Figure 12 refers to "radiative equilibrium with zero turbulent fluxes" and "moderately strong diffusive mixing". I could go on. I hope the author will see the need to adopt a concise and consistent terminology but also to consider whether a good scientific purpose is served by presenting results for cases in almost all figures. The terminological confusion is further accentuated on page 6 by having two variants to determine the skin temperature – no separate name is given to each case, and I am frankly not sure it is necessary to even present results from both, as the zero surface turbulent heat flux case is entirely theoretical.

We see the need to adopt a concise and consistent terminology.

We now define three cases: a "radiative case" with zero turbulent fluxes, a "radiative-convective case" with turbulent mixing in unstably stratified layers and a "radiative-diffusive case" with constant diffusivity in the troposphere and zero diffusivity in the stratosphere. --- And we changed the text at several places for consistency.

*5:15 "model includes a cirrus layer" – I think it is equally important to make clear that it ONLY includes a cirrus layer – i.e. no other cloud layers are included. The paper was not clear on this point but I regard this as a serious restriction when it comes to specifically looking at the impact of contrails, and so it is important that this is kept in mind. The impact of cirrus on the surface LW and SW budget, as well as the radiative heating at cirrus cloud base (e.g. Figure 7), will be considerably affected by the lower level clouds which are missing here.

We agree on the facts. The model code is prepared to include a liquid water cloud besides a cirrus cloud, and we tested it, but we now refrain from showing further results to reduce complexity. But we do mention that other clouds are important for the quantitative results.

*5:21 It is not clear where the value for diffusivity comes from. Some earlier study? The value plays such an important role in the analysis that it has to be justified in a more rigorous way. And it is important to again acknowledge important caveats: in this case, vertical heat transport in the real atmosphere is not, for the most part, diffusive, and so what is adopted here is a convenience for the simple model. C4 5:25-29 It is a little hard to follow this – given the signs shouldn't the "max" on line 26 be a "min"?

We now discuss the value for diffusivity in more detail, The turbulent flux $F_T$ is approximated as a function of a potential temperature gradient in the linearized form $dT/dz-\Gamma$, including the prescribed lapse rate $\Gamma$ and diffusivity $\kappa$ (Ramanathan and Coakley, 1978; Liou and Ou, 1983). The inclusion of $\Gamma$ makes sure that an atmosphere under threshold conditions with $dT/dz = -\Gamma$ experiences zero turbulent fluxes. The diffusivity $\kappa$ is set to zero in the stratosphere and to a constant $\kappa= 100$ m$^2$ s$^{-1}$ in the troposphere for simulation of diffusive mixing in this study. Liou and Ou (1983) used values up to 200 m$^2$ s$^{-1}$ to simulate cirrus in the tropical atmosphere. The diffusivity $\kappa$ causes vertical mixing in the troposphere with time scales $L_v^2/\kappa$ depending on vertical scales $L_v$ of temperature changes, about 10 d for mixing over the whole troposphere ($L_v \approx 10$ km) and about 3 h for a layer of 1 km depth. Stone (1973) estimates the effective diffusivity $\kappa_H$ for horizontal mixing by large-scale eddies to be at least $10^6$ m$^2$ s$^{-1}$. For similar time scales, the diffusivity $\kappa$ for vertical mixing should be related to $\kappa_H$ by the square of the ratio of vertical to horizontal length scales. The length scale ratio can be estimated from geostrophic equilibrium, $L_v/L_H \approx f^2/N^2$ where f and N are the Coriolis and the Brunt-Väisälä frequencies (Vallis, 2006). For typical mid-latitude and tropospheric values (f= $10^{-4}$ s$^{-1}$, N= 0.01 s$^{-1}$) one obtains $\kappa \approx (L_V/L_H)^2 \kappa_H \approx 100$ m$^2$ s$^{-1}$. These are of course only order of magnitude estimates.

*6:10 Using a surface albedo of 0.3 is a very crude way of mimicking low level clouds, and of course only does so in the SW (and so the LW surface budget is more sensitive to atmospheric perturbations than it would otherwise be). It is not quite clear to me why other clouds are excluded – is it an attempt to simplify or a methodological difficulty in including them? And why 0.3? I recognise this is the planetary albedo, but a surface albedo of 0.3 does not yield a planetary albedo of 0.3, because of atmospheric absorption (pushing one way) and Rayleigh scatter (pushing the other). It would be reassuring to know what the control top-of-atmosphere radiation budget is, as this would help determine how realistic the forcings (especially the longwave) are.

The albedo and SZA were initially selected because we also wanted to study cirrus effects globally. We now concentrate on mid-latitude values similar to previous contrail studies. The basic massage of the results is unchanged. We are now even more certain that our results are robust with respect to major changes.

In addition, we learned a lot from this exercise with respect to the quasi linear behavior of the model results and the different importance of SW and LW effects, and explained this in the revised text.

*6:10 "cos(SZA)=0.25" – this surprised me too. I understand that this yields the correct incoming solar radiation at top of atmosphere, but the high zenith angle (75 degrees) will significantly bias the SW effect of contrails to be more negative – indeed it is the zenith angle close to the most negative radiative forcing, according to the excellent Schumann et al. (2012 - 10.1175/JAMC-D-11-0242.1) paper and this may significantly affect some of the section 3.2 results . In radiative convective models (such as Manabe and Wetherald) it is common to assume a cos(SZA) of 0.5 and to assume a fractional day length of 0.5, although it may be more preferable to integrate over zenith angle.

Same response as above.

*7.10 Following on from the above comment, I am now a bit further confused. In the caption of Figure 2, it refers to the daily mean at 45N on 21 June. How does this relate to the cos(SZA)=0.25? And why is a surface albedo of 0.2 used here when it is 0.3 in the text? I guess Figure 2 is trying to justify the use of the 2-stream hexagons scheme used in the radiative-convective model, but it seems to me that it is not testing it for the conditions applied in that model. I am sorry if I misunderstand. And I have a similar query about Figure 3. Since, from my understanding, the radiative-convective model does not integrate over the diurnal cycle, this plot leaves a somewhat misleading impression and I am not sure of its purpose here. My bigger question is whether the C5 choice of cos(SZA)=0.25 leads to a bias in the SW budget of contrails. Also since a cirrus optical depth at 500 nm of 0.3 (10:25) is applied in the experiments, it is not clear why a value of 0.5 is used in Figure 3.

The confusion comes from the parameters used in Meerkötter et al. (1999). We now eliminated this discrepancy in using their values.

7:19 The figures show 360-720 ppb, the text says 300-600 ppb

We agree. But the figure is now deleted, as you suggested in the next comment.

7:17 Figure 4: While it is useful for the authors to have performed this calculation, I see no reason for including it in the paper – it is a result that is over 50 years old and in my view just inflates the paper. I feel something of the same way about Figures 5 and 6, since they are referred to only in passing. The inversion in Figure 6 may be something of an artefact resulting from the exclusion of lower level clouds

Figure 4 to 6 were shown to demonstrate that the code is able to compute the convective adjustment correctly. We now feel that convective adjustment is less important for this study. Hence, these figures are no longer necessary and we eliminated the figures and the corresponding sentences.

8:6 The expression for heating rate is textbook physics and doesn't need including – I am not sure the value for the lowest level is in any case correct, if the surface pressure is really 1013 hPa (I get 0.64 K/day).

We agree (though not all readers may have your knowledge), and we now reduced details.

8:12 "radiative EQUILIBRIUM case"

We agree and changed the text.

8:13 "smaller vertical scales" – it is hard to see this when the plot is presented in linear pressure.

We agree, and now deleted this argument - it is not necessary.

8:17: I agree that the 8-13 micron window is "more transparent" than neighbouring spectral regions, but it is hardly transparent, because of water vapour continuum absorption in this region.

We agree on the physics. We now changed the text to clarify this issue.

8:19 "stratosphere" – this sentence only makes sense to me if it is the "lower stratosphere"

We agree and changed accordingly.

8:21 "rather stable" – it is unclear what measure of stability is being used in making such a statement

We agree, and this text part was eliminated for shortness.

9:23 Perhaps 2 significant figures are enough in this and later paragraphs?

We agree and changed the numbers to 2 digits.

10:5-20 The experiment described here (100% cirrus, 150% perturbation to humidity) feels very contrived and in my view was a distraction. I suggest it be removed. C6

We follow the suggestion, and remove the strong cirrus and enhanced humidity cases from the figure. The results are still mentioned.

*10:24 Why 3% given the 0.2-0.5% at 2:7? But I am concerned that the assumed cirrus amount will ultimately impact the radiative heating in the upper troposphere and hence the extent to which that region can be decoupled from the atmosphere below. In addition, I suspect that the impact is also highly dependent on the height of the cirrus as well as the assumptions about underlying clouds.

The 3% is appropriate for mid-latitudes.

The introduction now says "Contrail cirrus clouds of significant optical thickness (>0.1) are estimated to cover about 0.2 - 0.5 % of the Earth, with higher values in northern mid-latitudes"

The cirrus amount is important for convective adjustment, but not for fixed diffusivity. This is no further explained in the text

Of course, the height of the cloud layer is important as are many other parameters. But the basic message, that mixing is important is independent of the specific parameter values. See revised manuscript.

11:1 "weak turbulent mixing" – which case is this referring to? See my comment **5. If you mean zero-mixing, the text should say this.

Yes.

11:6 "only for strong" – but as I understand you have only performed the experiment for zero or strong, so there is no intermediate case? I then get further confused by the discussion of convective mixing later in the paragraph, partly for the reasons discussed above, but partly because it is not shown on Figure 11. I suspect the result is also highly sensitive to the assumed cirrus height. Comparing Figure 5 and 6, it seems clear that convective mixing is impacting the temperature profile throughout the depth of the troposphere so it confused me to say that "convective mixing is weak"

We now revised the text considerably and hope that it is now clearer.

*11:10 The discussion at 6:10 about the chosen solar zenith angle calls into question this result, and I suggest it is revisited.

The case $Q_0 = 0$ is no longer discussed in detail (just mentioned), to reduce unnecessary complexity.

12:18 I can see no such plot in Ponater et al. (2006) – I am sorry if I miss it. Perhaps the text should refer to Figure 2 of Ponater et al. (2005) (see also 14:25) but even there I am a bit doubtful whether the point being made is the full story;

the maximum in upper tropospheric warming may be a result of well-known moist adiabatic processes (in which a surface perturbation is amplified at upper levels via the divergence of moist adiabats with height).

We intended to refer to Ponater et al. (2005).
The maximum in the upper troposphere cannot be explained by local release of latent heat, because the mass of water that condenses during cloud formation at those low-temperature levels is small. The pattern with enhanced temperature in the contrail region is also far more pronounced than in a similar $CO_2$ disturbance simulation (see Figure 1 of Ponater et al. (2006)).

The 2006 paper is a conference paper, available from http://elib.dlr.de/54467/. Here we show two essential panels from the Fig 1 in that paper (with permission by Michael Ponater).

[Figure]

Figure a: Zonal mean temperature response (in K) caused by contrails (RF = 0.19 W m$^{-2}$).

[Figure]

Figure b: Zonal mean temperature response (in K) caused by $CO_2$ disturbances (RF = 1 W m$^{-2}$).

13: I found this discussion rather conjectural and suggest it could be removed

We thought that the discussion should be of interest. Now we decided to remove chapter 4, and keep just a few sentences which are now in the newly formulated final Section "4 Summary, Implications and Conclusion".

*14: Although the central idea of this paper may indeed prove to be correct, this conclusions need to draw attention to the many caveats about the simple model that is adopted and how these may impact on the final result. C7

We agree, and revised the text. The conclusions now explicitly mention the model and parameter dependence and list arguments in support.

References cited here:

Liou, K.-N., and S.-C. S. Ou: Theory of equilibrium temperatures in radiative-turbulent atmospheres, J. Atmos. Sci., 40, 214-229, 1983.

Ponater, M., V. Grewe, R. Sausen, U. Schumann, S. Pechtl, E. J. Highwood, and N. Stuber: Climate sensitivity of radiative impacts from transport systems, in: Proceedings of an International Conference on Transport, Atmosphere and Climate (TAC), edited by: Sausen, R., Blum, A., Lee, D. S., and Brüning, C., University of Manchester and DLR Oberpfaffenhofen, http://elib.dlr.de/54467/, 190-196, 2006.

Ramanathan, V., and J. A. Coakley: Climate modeling through radiative-convective models, Rev. Geophys., 16, 465-489, 1978.

Stone, P. H.: The effect of large-scale eddies on climate change, J. Atmos. Sci., 30, 521-529, 1973.

Vallis, G. K.: Atmospheric and Oceanic Fluid Dynamics, Cambridge Univ. Press, Cambridge, 2006.

Ulrich Schumann and Bernhard Mayer 22 August 2017, changed version.

---

## Author Comment (AC3) · 22 Aug 2017

Responses to Reviewer 2.

We thank the reviewer for his thoughtful and detailed review. The review comments lead to considerable changes and several improvements.

The authors investigate the extent to which top-of-atmosphere forcing from jet contrails are able to influence the surface temperature using a simple radiative-convective diffusive model. This may be phrased as the "efficacy" associated with such forcing, and the authors show that this efficacy is strongly dependent on the assumed mixing within the model.

We agree. We now add "and efficacy relative to $CO_2$ changes" in the abstract to follow your interpretation.

The results of this study are of interest, but they are derived from a very simplified model, and because of its simplicity I am a bit unclear on the implications of this study for Earth's atmosphere. In particular:

We are pleased that the results are of interest.

We agree that the results are based on a simple model. That was the purpose.

As you know, our team also runs more comprehensive climate models. The problem is that such models often do not allow identifying reasons for certain results. Therefore, we looked by purpose for the most simple model we could think off to study the relative importance of mixing and radiation in clear isolation from other processes. In the conclusions, the importance of the model simplifications is now stressed. Further the abstract says: "Since the results of this study are model dependent, they should be tested with a comprehensive climate model in the future. "

1) The authors find that in the limit of weak tropospheric vertical mixing, the effect of upper tropospheric forcing like that of contrails can be to cool the surface. This seems to run counter to GCM studies of Hansen et al. (2006) and Ponater et al. (2006), which show a more constant tropospheric response, presumably because they have some vertical mixing. Does this mean that the weak vertical diffusion case in this study is simply not relevant to Earth's atmosphere?

We agree that the limiting result cannot be guaranteed to be fully relevant for real atmospheres and we now say this in the conclusions. But as noted in the introduction recent research indicate that strong SW contributions are getting more and more realistic. Certainly, this needs further studies and this paper may trigger such studies.

2) In the mid-latitude case, convection is hardly active because the large-scale forcing Q_0 stabilises the atmosphere. But in Earth's atmosphere, convection acts intermittently and the convective mixing is therefore underestimated by this model. Further, I think it is unreasonable to expect Q_0 to remain unchanged in response to the forcing. The thermal stratification of the midlatitudes is set by this large-scale forcing, and a change in this thermal stratification will likely have an influence on the midlatitude eddies. Is the vertical diffusion meant to be a parameterisation of these missing processes? If so, what level of vertical diffusion is relevant for Earth's atmosphere?

The reviewer addresses important issues, which we cannot answer strictly without running far more extensive models. Our point should still be valid that mixing is important. The question whether our study gives correct quantitative result cannot be answered without further research. We now say this in the abstract and in the conclusions.

3) In the "tropical" case ($Q_0 = 0$; Fig. 6) the convective adjustment is controlling the lapse rate, as is the case in Earth's tropics. But here, the forcing applied is very strong: 100% Cirrus cover. In this case, the Cirrus produces an inversion in the upper troposphere, and drives a second convective cell above the tropopause. I'm not sure this is a plausible outcome of contrail forcing. What happens if the forcing is reduced to a cloud cover of 0.2-0.5%? Do you still get the same decoupling from the surface? How does this depend on the height of the forcing?

You are right that the cirrus cover is important for stabilization and we mentioned that. We now deleted this part to reduce the complexity of the paper.

4) What does Fig 11 look like with radiative and convective adjustment? we expect the CO2 response to warming to be relatively uniform in the troposphere. This is true with high diffusion, but does not seem to be true in the radiative-convective case. To me this suggests that the no diffusion limit is not relevant for the Earth.

We now show the results also for convective adjustment. This discussion is part of the discussion on model dependence. We now point out that global models often show a rather smooth profile of temperature increase in the troposphere, partly perhaps because of strong mixing on coarse grids.

My suggestions to improve the manuscript in light of these comments are as follows:

Thank you for your suggestions. We revised the paper accordingly.

- More consistent forcing levels across the experiments. In some cases the Cirrus cloud cover is set to 3%, in others 100%. Why is this the case? And how was 3% chosen? It seems much larger than the 0.2-0.5% quoted in the introduction for Contrail fraction. Does the response depend on the size of the forcing? What about the height of the forcing?

We keep less cases, with 3% cover (the 100 % case is kept in the comparison to Meerkötter et al (1999) who run the test cases with 100 % contrail cirrus cover). We discuss the importance of cirrus properties. The strength of the cirrus forcing is important mainly for convective mixing. The diffusive mixing is linear in this model and less sensitive to the contrail details. The main issue that contrails have positive RF at TOA and negative RF at the surface is robust and independent of such details.

- Some more discussion on how the results from the simple model should be interpreted. In particular, what is the level of vertical mixing relevant for Earth's atmosphere in midlatitudes and in the tropics? How does the assumption of diffusive mixing affect the results.

We agree. We now relate the diffusivity to the studies by Stone on baroclinic adjustment by large scale eddies.

- I think the study would benefit from using single-column model with a more realistic description of convection than the simple model used here (e.g., the single column model of a IPCC-class GCM). While this does not ameliorate all the problems with using a 1-D description of the atmosphere, it will ensure the convective response given the mean state will be somewhat realistic, particularly for the "tropical" case in which Q_0 is zero.

We do not follow this suggestion because we will never find a 1-d model that includes all known effects. (A future model version should be 2-dimensional and include diural and seasonal cycles – coming closer to a full climate model.) Instead, by purpose, we simplify the study even further and skip some results of variants for clarity The test against reality has to be done within comprehensive climate models. We say this in the Conclusions. However, we show and explain the robustness of the results to parameter changes.

Minor comments:

page 1: Line 8: What does "basically without climate system changes" mean? Does this refer to the dynamic heating in the model? This should be clarified here and in the other places where this statement is made in the manuscript.

The term "basically" was used since we had a model variant with fixed relative humidity. But we now skip this and simplify the paper.

page 2: line 33: Here it is argued that contrails do not behave the same as high clouds, but later the forcings you apply are described as either thin cirrus or contrails. This contradiction should be resolved

We thought that our study is relevant beyond contrail cirrus. That caused part of the complexity and apparently misleading wording. We now decided to reduce the paper to the mid-latitude case and talk about contrail cirrus only (with a short remark on generalization potentials in the Conclusions).

page 3: Line 32: I am not sure what it means to avoid warming contrails. Does this mean that one mitigation option is to move flight paths to regions in which the effects of contrails is a cooling?

Your are right in your interpretation. We now added "route changes" to clarify this question.

page 5: Line 1-10: The discussion here is very confusing. At one point it is stated that Q_0 is the sum of the divergence of F_R and F_T, but it is a bit unclear whether this statement is supposed to only apply for T = T_0 or more generally. Later it is stated that the Q_0 = 0 case is "pure radiative equilibrium", but I think this should be Q_0 = 0 and F_T = 0.

We now changed the text to avoid such misunderstandings.

page 5: line 20: I don't understand why \Gamma drops out of the equation for \Delta T, or why the contribution from \Gamma affects Q_0. Isn't Q_0 fixed? I think the equation for \Delta T should be presented for clarity.

The reference lapse rate $\Gamma$ drops out for constant diffusivity. This can be seen when taking the difference of Eq. (1) for $\Delta T = T(t,z) - T_0(z)$. The corresponding equation for $\Delta T$ would make the text more lengthy without providing much insight. Basically this is of theoretical importance only. The code includes the full set of equations. Therefore, we now deleted this sentence.

page 6: line 10: Setting the cosine zenith angle to 1/4 biases the solar radiation to have a high zenith angle, this will increase the reflection from clouds and bias the results. For the global mean, one should use the insolation weighted zenith angle (Cronin 2014). But I do not see why the global mean insolation is necessarily desired. The temperature profile used is one of the mid-latitudes, so presumably that is the focus. Why not use a diurnally varying solar insolation for e.g., 45 deg?

We changed the values to mid-latitude values. The results are robust to these changes.

page 6: line 20: The radiation only boundary condition for T_skin is unphysical for cases with turbulent fluxes. Perhaps it would make more sense to use an assumed value of the surface enthalpy exchange coefficient and wind speed that are typical of Earth's surface conditions.

We decided to reduce complexity by setting the surface temperature equal to the temperature in the lowest model layer, throughout the paper. Again, a more realistic model would require further model parameters, which we want to avoid, because any parameter requires a discussion on its validity and limitation and this would make the paper just more complex without much gain and without changing in the basic conclusions.

page 8: line 32: The Hansen et al. (1997) result needs explaining. What type of model were they using? Does this indicate that the strong mixing limit is the appropriate one?

We now explain that Hansen et al. (1997) used a GCM with rather coarse resolution.

page 10: line 25: Here 3% Cirrus coverage is used, but the global cover mentioned in the introduction is 0.2-0.5%. Does the magnitude of the Cirrus cover have any effect on the results?

We use 3 % because that is representative for mid-latitudes. We now explain this.

page 11: line 9: It appears that the Cirrus drives convection above it to the tropopause. Is this likely for the forcing from Jet contrails in the next century?

It is well known that the radiative heating in a cloud layer may drive convection above it, and this is what the model simulates. This does not mean that all contrail cirrus cause convection, and we do not say that. The text got modified for avoid this misunderstanding.

Ulrich Schumann and Bernhard Mayer, 22 August 2017

---

## Author Response (AR2)

Dear Editor,

we thank you for your decision on our paper(acp-2017-465).

We have performed a minor revision as requested.

We carefully considered the new comments by Reviewer #1. The comments caused a few changed and technical corrections. We followed his suggestions as far as we think it is reasonable. See our detailed point-by-point response.

We gladly noted that Reviewer #2 agreed to the publication

We hope that the paper is now acceptable for ACP.

Best regards,

Ulrich Schumann and Bernhard Mayer

6 October 2017

Reviewer comments in black. Authors response in color.

We thank the reviewer for his careful reading and thoughtful comments.

We changed the text and a figure accordingly

1. I was encouraged to read in their response to my initial comments that the authors had "deleted … the pure radiative equilibrium case". I was disappointed to find that they had not done so. It features prominently in the figures and in Table 1, which is the main results table. I maintain my view that results from the pure radiative case are of little relevance to any real world cases, and the authors have not defended its retention, but continue to highlight the results from this case in, for example, the abstract.

In revision, we had indeed deleted the "pure radiative equilibrium case", which is a case without diffusive mixing and with zero dynamical heating (as represented in our former figures 4, 5 and 6). We have not deleted, however, the "radiative case" with zero mixing and fixed dynamical heating which is needed because the differences between the "radiative-diffusive case" and the "radiative case" are taken as measure of the importance of mixing. The results for the "radiative convective" case (with fixed dynamical heating) were additionally included in Figure 8 also because of a corresponding request of Reviewer #2. This "radiative, convective" case is not included in Table 1, because of the small differences to the "radiative diffusive" case, as was explained in the text.  Hence, this comment leaves the paper unchanged.

2. My original comment *5:15 seems to have been ignored. The authors say they "agree on the facts" but the revised text does not make clear that the model includes ONLY a cirrus layer, and no other clouds. (and I apologise if I miss it, but the authors have not signposted where they have done so in their response)

The comment has not been ignored, but we have not discussed studies with a liquid layer cloud (which we had performed) in order not to complicate the paper. To clarify, we now changed the introduction and write: "Other clouds and aerosols  are not included in this study."

3. In addition, I consider my comments on *7:10 and 8:6 to have been effectively ignored. In response to *7:10 the authors say they "have eliminated this discrepancy" but I do not know what this means, as the Figures in question remain in the manuscript. For 8:6 I had a specific comment on a value in the text which has neither been challenged nor corrected and my contention that the text is incorrect remains. The response to 12:18 is also somewhat unsatisfactory. The upper tropospheric amplification in the CO2 case is not the result of condensation at low temperatures, but due to the divergence of moist adiabats at lower altitudes. The same can also be true for the contrail case.

The former point *7:10 questioned the assumptions for SZA and albedo values used.

We did not ignore this comment. In fact, we changed the values for cos(SZA) from 0.5 to 0.6, now with daytime fraction 0.64, to come closer to mid-latitude summer mean values, and justified the use of the selected albedo value 0.2 by reference to former studies with the same value.

Any change of SZA and albedo changes the cirrus results in quantitative details. But the changes have small effects because the imposed fixed dynamical heating changes accordingly. Changes in SZA and albedo have zero impact on the ghost forcing results because that is a pure LW forcing. Hence, our main conclusion that mixing is important for efficacy, is fully independent of the SZA/albedo values assumed. Hence, this comment leaves the paper unchanged.

The former point 8.7 questioned the numerical value of the heating rate in the lowest model layer. That point was well posed. We had to correct this value in the write-up and we did so. Hence, again, this point was not ignored.

With respect to 12:18, we wa agree other reasons and, hence, we cannot exclude that "the divergence of moist adiabats at lower altitudes" may explain differences in the $CO_2$ and contrail responses, but we think that our wording in Section 3.2 is cautious enough ("the mixing was **likely** not strong

enough to disperse the contrail-induced radiative heating uniformly over the troposphere.") Nevertheless, we now cautioned the Conclusion: Instead of "One climate model study (Ponater et al., 2005) indicates …" , we write "The results of one climate model study (Ponater et al., 2005) support …".

4. At line 298, Figure 8 has serious problems and has gone backward since the earlier version (when it was Figure 11). The right-hand side of the figure purports to show the instantaneous heating rate change due to CO2, but in fact the red lines are the contrail heating rates repeated from the left hand side. To add to the confusion the caption says that the left hand side is CO2 and the right hand side is contrails, when it is the other way round.

We agree and thank the reviewer for his remarks.  There was indeed a mistake which occurred when we combined the various panels form the former figure into one figure during the revision. It is now corrected. Also the caption (left/right) is now corrected.

5. The discussion surrounding Table 1, the main results table, is confusing. As far as I can tell, the values in the table have changed (due to the change in insolation parameters) but the text is essentially unchanged and now inconsistent. At line 326, it says the contrail RFa is small, but it is no longer small, but 0.42 W m-2, and so a substantial fraction of the CO2 value (0.72). The same is reported at line 400 and again at line 408 where the SW and LW forcings are said to be "nearly cancelling" where they are not. The LW forcing is almost double the SW forcing now, and the cancellation is only partial. In the radiative-diffusion case, the climate sensitivity for the separate LW and SW cases is 0.24 and 0.27 respectively. So while the authors write at lines 333 and 334 that SW efficacy is larger than the LW efficacy, and this is of course correct, perhaps it is as notable that they are so similar, given the different signs of the forcing and the different surface atmosphere partitioning. At line 405, presumably RFi should be RFa (as efficacies for the instantaneous case are never shown) and the following text says there are "strong" departures from unity. But they are not actually that strong (the efficacy is about 0.75). Part of my point here is that the authors could easily

be quantitative in the text – "strong departures" is a matter of opinion, and citing the actual values would provide a better perspective for the reader to judge.

We thank the reviewer for these remarks. The values in the table had been changed for the reasons mentioned, but the text did not fully reflect these changes. This is now corrected.

The new text now includes corresponding changes in discussing Table 1 in Section 3.2 and corresponding changes in the Conclusions.

6. The two sentences at line 402 are misleading. The "zero mixing case" (by which the authors mean the radiation-only case) is not a credible real-world case and hence the deduction from this case that the temperature sensitivity could be negative is beyond speculative, on the evidence presented. This issue is also present in the abstract at line 20, where it refers to a "low mixing case", even though the text only ever shows that a cooling occurs for a "no mixing case". I am not sure why the abstract doesn't focus on the diffusive case, which is more realistic. One could easily interpret the results as showing that the difference in efficacies between the CO2 and contrail cases are really rather small, given the quite different nature of the forcings, and perhaps by being quantitative in the abstract it would allow the readers to make their own minds up, instead of being left with a potentially misleading impression.

These are again valuable remarks.

Please note that the Conclusion already contained the following sentences: "Hence, though our study shows the principle importance of mixing for climate sensitivity to contrails, we cannot say how important mixing is for real world cases quantitatively. Ultimately, this requires careful simulations with a comprehensive climate model."

Nevertheless, we changed the Conclusion section slightly in the sense as suggested: We changed and added two sentences:

"Hence, in such an extreme case, the temperature sensitivity could be negative. In the analyzed case without mixing, the LW and SW $\lambda_a$ values differ by a factor of 4.4 but only by 10 % with diffusive mixing."

In the abstract, the "For low mixing conditions, …" has been replaced by "For zero mixing, …" to make clear that this is a theoretical limiting case: We think that we have addressed the reviewers's concerns this way.

7. In passing I note that the new text focuses on the mid-latitude summer continental case, while also maintaining that the timescales for mixing are determined by large-scale eddies. I have struggled to find anything quantitative to challenge this statement, but some care is needed. During summer in mid-latitude continental regions, my guess is that convective heat transport is at least competitive with the vertical heat transport from large scale eddies, and may even be larger.

For the present study, any details of how the mixing is accomplished are of minor importance. Important are the differences between cases without and with mixing. We agree: Convective heat transport may be competitive with the vertical heat transport from large scale eddies. But to quantify the relative importance of various mixing mechanisms would requires a climate model including all these transports. In a sense, the reviewer now supports our conclusion that mixing is important for climate sensitivity. – No text change.

[revised manuscript text omitted]